EMBO
Molecular Medicine

# Single-domain antibodies targeting antithrombin reduce bleeding in hemophilic mice with or without inhibitors

Elena Barbon[1],[†] (ID), Gabriel Ayme[2],[†], Amel Mohamadi[2], Jean-François Ottavi[3], Charlotte Kawecki[2], Caterina Casari[2], Sebastien Verhenne[2], Solenne Marmier[1], Laetitia van Wittenberghe[1], Severine Charles[1], Fanny Collaud[1], Cecile V Denis[2], Olivier D Christophe[2], Federico Mingozzi[1],[*],[‡] (ID) & Peter J Lenting[2],[‡],[**] (ID)

## Abstract

**Novel therapies for hemophilia, including non-factor replacement and *in vivo* gene therapy, are showing promising results in the clinic, including for patients having a history of inhibitor development. Here, we propose a novel therapeutic approach for hemophilia based on llama-derived single-domain antibody fragments (sdAbs) able to restore hemostasis by inhibiting the antithrombin (AT) anticoagulant pathway. We demonstrated that sdAbs engineered in multivalent conformations were able to block efficiently AT activity *in vitro*, restoring the thrombin generation potential in FVIII-deficient plasma. When delivered as a protein to hemophilia A mice, a selected bi-paratopic sdAb significantly reduced the blood loss in a model of acute bleeding injury. We then packaged this sdAb in a hepatotropic AAV8 vector and tested its safety and efficacy profile in hemophilic mouse models. We show that the long-term expression of the bi-paratopic sdAb in the liver is safe and poorly immunogenic, and results in sustained correction of the bleeding phenotype in hemophilia A and B mice, even in the presence of inhibitory antibodies to the therapeutic clotting factor.**

**Keywords** anticoagulation; coagulation; gene therapy; hemophilia; single-domain antibodies
**Subject Categories** Haematology; Immunology

See also: **JM O'Sullivan & JS O'Donnell** (April 2020)

## Introduction

Hemostasis is a complex process dedicated to limit blood loss upon vascular injury. In this process, it is the task of the coagulation cascade to generate a fibrin network that is required to stabilize the platelet plug (Versteeg *et al*, 2013). The coagulation pathway comprises a series of sequential steps in which pro-enzymes are converted into active serine proteases (Davie *et al*, 1991). Ultimately, this cascade leads to the formation of thrombin, the enzyme that is responsible to convert fibrinogen into fibrin. Complementary to the proteolytic steps that activate the coagulation enzymes, there is also the need to have feedback regulatory loops that dampen coagulation (Versteeg *et al*, 2013). Accordingly, several anticoagulant proteins exist that are capable of blocking the function of coagulation enzymes or their cofactors. The fine balance between pro- and anticoagulant factors is in fact needed to prevent hypo- or hyper-coagulative states that, in some conditions, can be pathologic. Hemophilia is a well-known example of inherited alteration of the hemostatic balance due to a coagulation factor deficiency, which results in a bleeding diathesis. The lack of the procoagulant protein factor VIII (FVIII) for hemophilia A or the serine protease-precursor factor IX (FIX) for hemophilia B affects 1–2 *per* 10,000 and 1 *per* 25,000 males at birth, respectively (Bolton-Maggs & Pasi, 2003). Hemophilia A and hemophilia B are clinically indistinguishable, and their treatment is dictated by the clinical severity. Bleedings are usually efficiently prevented or resolved via replacement therapy using plasma-derived or recombinant factor concentrates (Manco-Johnson *et al*, 2017). Nevertheless, protein replacement therapy has a number of limitations, including the necessity for relatively frequent intravenous access and the development of neutralizing inhibitors in up to 30% of patients in hemophilia A and about 3% in hemophilia B patients (Eckhardt *et al*, 2013; Gouw *et al*, 2013; Calvez *et al*, 2014; Peyvandi

1 Genethon, Institut National de la Santé et de la Recherche Médicale U951 Integrare, Université Paris-Saclay, University of Evry, Evry, France
2 HITh, UMR_S1176, Institut National de la Santé et de la Recherche Médicale, Université Paris-Saclay, Le Kremlin-Bicêtre, France
3 Inovarion, Paris, France
*Corresponding author. Tel: +1 215 282 0134; E-mail: federico.mingozzi@sparktx.com
**Corresponding author. Tel: +33 149595651; Fax: +33 146719472; E-mail: peter.lenting@inserm.fr
†These authors contributed equally to this work as first authors
‡These authors contributed equally to this work as senior authors

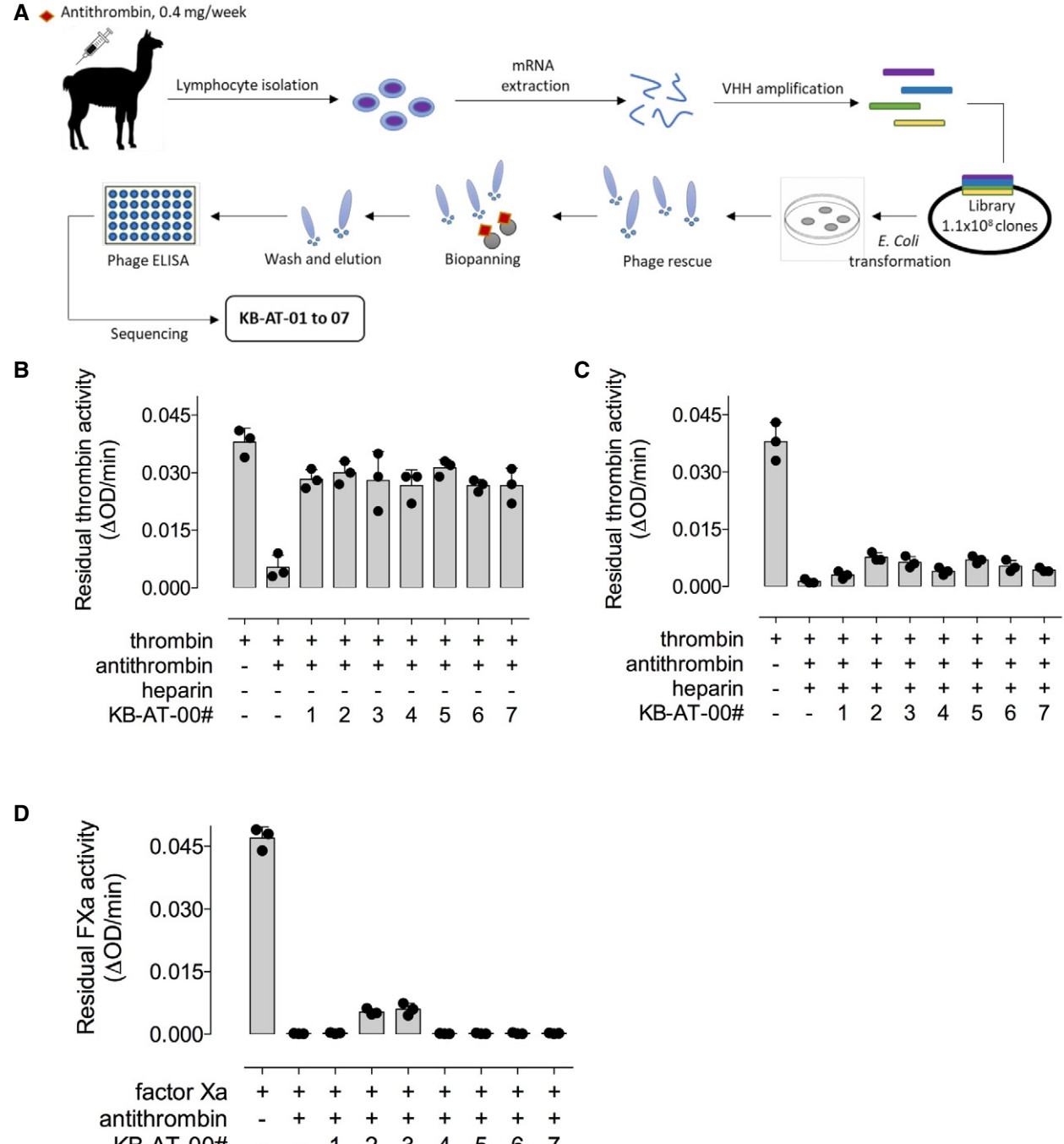

**Figure 1. Selection of anti-antithrombin single-domain antibodies and test of their inhibitory activity *in vitro*.**

A Schematic representation of the sdAB library selection and identification of monovalent KB-AT-01 to KB-AT-07 antibodies.

B Bar graph representing residual thrombin activity (ΔOD/min) in the presence of the sdAbs without heparin.

C Bar graph representing residual thrombin activity (ΔOD/min) in the presence of the sdAbs and heparin.

D Bar graph representing residual activated factor X (FXa) activity (ΔOD/min) in the presence of the sdAbs and heparin.

Data information: Data represent the mean ± SD of three experiments.

*et al*, 2016). As an alternative to protein infusion, gene-based strategies to address the clotting factor deficiency in hemophilia are being explored. Among these, liver gene transfer with adeno-associated virus (AAV) vectors is now in early- and late-stage clinical

development (Nathwani *et al*, 2014, 2017; George *et al*, 2017; Miesbach *et al*, 2018).

Despite the advances in the treatment of hemophilia, patients with inhibitors had only limited therapeutic options until recently.

**Table 1. Recognition of antithrombin in plasma by sdAbs.**

| | Human | Simian | Mouse | Rat | Rabbit | Canine | Bovine | Porcine |
|---|---|---|---|---|---|---|---|---|
| KB-AT-01 | ++ | ++ | ++ | + | + | − | − | − |
| KB-AT-02 | ++ | ++ | ++ | ++ | ++ | ++ | ++ | ++ |
| KB-AT-03 | ++ | ++ | ++ | + | − | + | − | − |
| KB-AT-04 | + | + | ++ | + | − | + | − | + |
| KB-AT-05 | + | + | ++ | − | − | − | − | − |
| KB-AT-06 | ++ | ++ | + | + | ++ | + | − | + |
| KB-AT-07 | ++ | ++ | + | − | − | − | − | − |
| Positive cntl | ++ | ++ | ++ | ++ | + | ++ | + | ++ |
| Negative cntl | − | − | − | − | − | − | − | − |

Antibodies immobilized at 10 μg/ml, and plasmas diluted fourfold; Positive cntl: positive control, polyclonal anti-antithrombin antibodies (Affinity Biologicals). Negative cntl: no antibody immobilized. Symbols represent the following: Negative binding defined as OD being ≤ 0.1; +: Moderate positive binding defined as OD being > 0.1 − < 0.5; ++: Strongly positive binding defined as OD being ≥ 0.5.

However, novel treatment strategies not relying on factor replacement are now resulting in important clinical advances (Muczynski et al, 2017). Among these, one potentially promising approach is to restore the hemostatic balance by neutralizing components of the anticoagulant pathway, such as tissue factor pathway inhibitors, activated protein C, or antithrombin (Sehgal et al, 2015; Polderdijk et al, 2017; Chowdary, 2018). Indeed, current late-stage trials are evaluating the possibility to use siRNA-mediated silencing of antithrombin expression (which results in reduced levels of circulating antithrombin) as a mean to correct the bleeding diathesis in hemophilic patients (Sehgal et al, 2015; Pasi et al, 2017). In an alternative strategy aimed at targeting the antithrombin pathway and restore hemostasis in hemophilia, here we describe the development of recombinant single-domain antibodies (sdAbs, also known as heavy chain-only antibodies or nanobodies) binding antithrombin. We show that by combining sdAb monomers selected for binding antithrombin, it is possible to achieve profound inhibition of antithrombin activity. Accordingly, sdAbs against antithrombin were able to restore thrombin generation in vitro. Delivery of bi-paratopic sdAbs, both as recombinant protein or as engineered to be expressed from the hepatocytes via AAV vector-mediated gene transfer, significantly reduced blood loss in hemophilic mice, even in the presence of inhibitory antibodies against the clotting factor.

## Results

### Library generation and selection of antithrombin-specific monovalent sdAbs

To generate a sdAb library, a llama was immunized via subcutaneous injection of 0.4 mg of human antithrombin, once weekly during 4 weeks (Fig 1A). Blood was then collected to isolate total mRNA from B lymphocytes, which was used to generate a library of heavy chain-only encoding cDNA. This library consisted of $1.1 \times 10^8$ independent clones. The quality of the library was assessed by analyzing the sequence of 200 randomly picked clones, revealing that each clone contained a unique sequence. Protein expression analysis of these clones demonstrated that 182 (92.5%) of this subset of clones expressed sdAb proteins with the expected size of 13–15 kDa. The full library was then used to select

antithrombin-specific sdAb via phage-display screening using beads coated with purified plasma-derived human antithrombin (0.2 mg/200 μl beads). After two rounds of panning, 7 unique clones were selected for further analysis (Fig 1A).

### Monovalent sdAbs inhibit antithrombin activity in the absence of heparin

Seven sdAbs targeting antithrombin were expressed as histidine (HIS)-tagged monovalent proteins in E. coli-SHuffle cells and purified to homogeneity via $Co^{2+}$-affinity and gel filtration chromatography. Purified sdAbs were tested for cross-reactivity with antithrombin from different species present in fourfold diluted plasma. All seven sdAbs (designated KB-AT-01 to -07) did recognize antithrombin from multiple species, including human and murine antithrombin, to a variable degree (Table 1). A dose response for the binding of purified human and murine antithrombin to immobilized sdAbs is presented in Fig EV1. We then evaluated the neutralizing potential of monovalent sdAbs toward antithrombin in the absence and presence of heparin, which enhances the inhibitory potential of antithrombin toward thrombin. In the absence of heparin, all monovalent sdAbs partially neutralized antithrombin-mediated inhibition of thrombin (range: 55–67%; Fig 1B). In contrast, when heparin was added to the reaction mixture, the sdAbs lost their capacity to neutralize antithrombin (Fig 1C). A similar lack of neutralizing activity of the sdAbs was observed for the inhibition of factor Xa (FXa) in the presence of low-molecular-weight heparin (Fig 1D). These results suggest that, in their monovalent conformation, sdAbs lack enough potency to block the activity of heparin-bound antithrombin.

### The bi-paratopic sdAb KB-AT-23 is a potent inhibitor of antithrombin activity even in the presence of heparin

To overcome the limited ability of monovalent sdAbs to neutralize antithrombin in the presence of heparin, several bi- and multi-paratopic variants were designed using combinations of different monovalent sdAbs. After a preliminary selection based on neutralizing capacity of the multivalent sdAbs in the presence of heparin, an in-depth analysis was performed with the candidate KB-AT-23 (consisting of sdAbs KB-AT-02 and KB-AT-03, separated

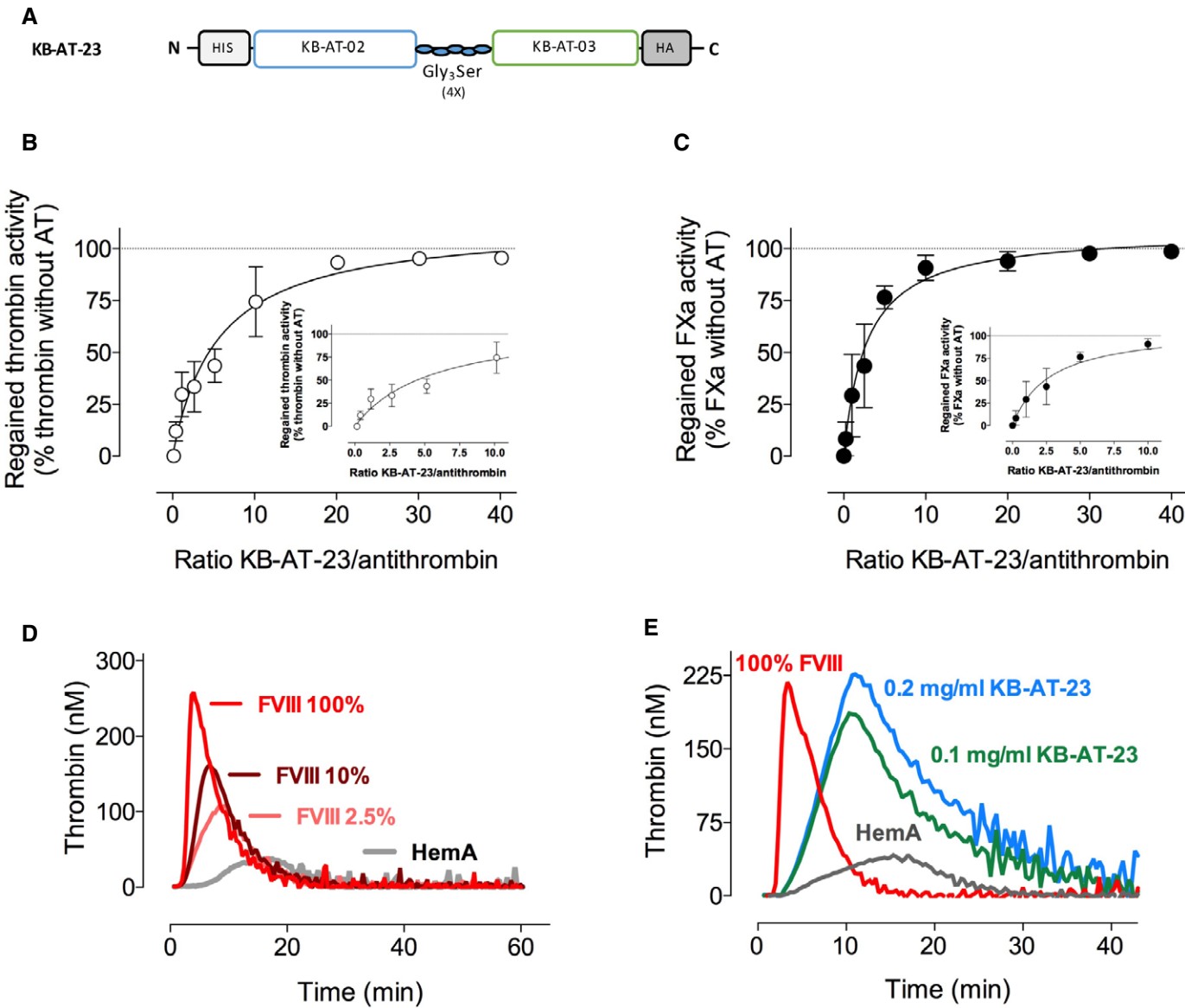

**Figure 2. Engineering of a bivalent KB-AT-23 sdAb and test of its antithrombin inhibitory activity *in vitro*.**

A   Schematic representation of the bivalent KB-AT-23 sdAb.
B, C   Graphs reporting the recovery of thrombin- and FXa-mediated substrate conversion in the presence of increasing doses of KB-AT-23, in the presence of antithrombin and heparin. Data represent mean $\pm$ SD of three experiments.
D, E   Thrombin generation profiles of FVIII-deficient plasma spiked with different concentrations of recombinant FVIII or 2 doses of KB-AT-23. Data are representative thrombograms. Detailed information on thrombin generation parameters is provided in Table 2. Antithrombin activity was inhibited by 81.1 $\pm$ 6.1% and 54.1 $\pm$ 8.4% for the 0.2 and 0.1 mg/ml concentration, respectively.

by a triple Gly$_3$Ser-linker; Fig 2A). KB-AT-2 and KB-AT-3 can bind simultaneously to antithrombin, indicating that they recognize separate regions within the antithrombin molecule. The combination of these sdAbs was chosen since the monovalent variants showed the highest residual neutralizing potential in the presence of heparin (Fig 1C and D). KB-AT-23 dose dependently interfered with antithrombin activity, as illustrated by the recovery of thrombin- and FXa-mediated substrate conversion in the presence of antithrombin and heparin (Fig 2B and C). The IC$_{50}$ for thrombin-neutralizing inhibition by antithrombin was achieved at a

molar sdAb/antithrombin ratio of 5.5 $\pm$ 0.9 (Fig 2B). For FXa, the IC$_{50}$ was obtained at a molar ratio of 2.8 $\pm$ 0.5 (Fig 2C), indicating that KB-AT-23 is more efficient in neutralizing antithrombin activity toward FXa compared to thrombin. In a next series of experiments, the ability of KB-AT-23 to restore thrombin generation in FVIII-deficient plasma was evaluated. For comparison, a thrombin generation profile of FVIII-deficient plasma spiked with recombinant FVIII to a concentration of 2.5% (0.025 U/ml; representing moderate hemophilia A), 10% (0.1 U/ml; representing mild hemophilia A), or 100% (1 U/ml;

Table 2. Thrombin generation test in human FVIII-deficient plasma.

| Added component in FVIII-deficient plasma | N | ETP (μM•min) | Thrombin Peak (nM) | Lag time (min) | Time to Peak (min) |
|---|---|---|---|---|---|
| FVIII (1 U/ml) | 4 | 1.7 ± 0.1 | 262 ± 23 | 2.8 ± 0.3 | 4 ± 1 |
| FVIII (0.1 U/ml) | 3 | 1.3 ± 0.1 | 163 ± 14 | 3.3 ± 1.2 | 7 ± 2 |
| FVIII (0.025 U/ml) | 3 | 1.0 ± 0.1 | 106 ± 11 | 3.4 ± 1.4 | 11 ± 3 |
| None | 5 | 0.3 ± 0.1 | 30 ± 3 | 5.8 ± 1.9 | 15 ± 5 |
| KB-AT-0203 (3.3 μM) | 3 | 2.7 ± 0.5 | 185 ± 11 | 3.3 ± 0.6 | 9 ± 3 |
| KB-AT-0203 (6.6 μM) | 3 | 3.6 ± 0.7 | 225 ± 23 | 3.8 ± 0.7 | 8 ± 2 |
| Normal plasma | 6 | 1.5 ± 0.1 | 310 ± 5 | 3.0 ± 0.3 | 5 ± 1 |

Parameters for measuring thrombin generation (ETP, thrombin peak, lag time, and time to peak) were measured in immune-depleted FVIII-deficient human plasma in the presence of tissue factor (1 pM) and phospholipids (4 μM) with or without FVIII, or with KB-AT-23. Data are presented as mean ± SD.

Table 3. Thrombin generation test in murine FVIII-deficient plasma.

| Added component in FVIII-deficient plasma | N | ETP (μM•min) | Thrombin Peak (nM) | Lag time (min) | Time to Peak (min) |
|---|---|---|---|---|---|
| None | 4 | 0.3 ± 0.1 | 35 ± 11 | 0.8 ± 0.3 | 3.3 ± 0.3 |
| KB-AT-23 (2 μM) | 4 | 0.9 ± 0.1 | 121 ± 7 | 0.8 ± 0.2 | 2.6 ± 0.2 |
| KB-AT-23 (2 μM) + Anti-FVIII (8 BU/ml) | 4 | 0.9 ± 0.1 | 111 ± 6 | 0.5 ± 0.2 | 2.0 ± 0.1 |
| KB-AT-23 (2 μM) + Antithrombin (2 μM) | 4 | 0.2 ± 0.1 | 31 ± 3 | 0.6 ± 0.2 | 2.7 ± 0.3 |
| Wild-type plasma | 3 | 1.1 ± 0.1 | 147 ± 9 | 0.9 ± 0.2 | 2.6 ± 0.2 |

Parameters for measuring thrombin generation (ETP, thrombin peak, lag time, and time to peak) were measured in murine FVIII-deficient human plasma in the presence of tissue factor (0.75 pM) and phospholipids (4 μM). Data are presented as mean ± SD.

representing normal plasma) was generated (Fig 2D). As expected, the addition of FVIII to FVIII-deficient plasma resulted in a dose-dependent increase in thrombin generation (Fig 2D). All parameters like endogenous thrombin potential (ETP), lag time, and thrombin peak increased in a FVIII-dependent fashion (Table 2). The addition of 0.1 mg/ml (3.3 μM) or 0.2 mg/ml (6.6 μM) of KB-AT-23 also enhanced thrombin generation in a dose-dependent manner (Fig 2E). Although time to peak was longer upon the addition of KB-AT-23, the amount of thrombin generated was markedly increased when compared to normal plasma or to FVIII-deficient plasma supplemented with FVIII to 1 U/ml (Table 2). We further analyzed the effect of KB-AT-23 in plasma from FVIII-deficient mice. Thrombin generation in murine FVIII-deficient plasma was 0.3 ± 0.1 μM•min compared to 1.1 ± 0.1 μM•min for wild-type plasma (Table 3). The presence of KB-AT-23 (2 μM) increased the endogenous thrombin potential (ETP) in murine FVIII-deficient plasma to 0.9 ± 0.1 μM•min ($P = 0.158$ compared to wild-type plasma; $P < 0.0001$ compared to FVIII-deficient plasma). Interestingly, the effect of KB-AT-23 was unaffected when anti-FVIII antibodies (titer 8 BU/ml) were present (ETP = 0.9 ± 0.1 μM•min; $P = 0.998$ versus KB-AT-23 alone). In contrast, the KB-AT-23-induced increased thrombin generation could be annulated by the addition of antithrombin concentrate (2 μM; ETP = 0.2 ± 0.1 μM•min: $P = 0.260$ compared to FVIII-deficient plasma). Together, these results suggest that combining two distinct sdAbs, each having limited neutralizing potential, into a single molecule increases its neutralizing capacity, allowing the restoration of thrombin generation in FVIII-deficient plasma. In addition, its effect is unaffected by anti-FVIII inhibitory antibodies and can be reversed by the addition of antithrombin concentrate.

## KB-AT-23 administration restores hemostatic balance in hemophilia A mice

We next explored the *in vivo* characteristics of KB-AT-23. To determine its *in vivo* circulatory survival (Fig 3A), we generated two distinct sdAb-fusion proteins. One consisting of KB-AT-23 fused to von Willebrand factor (VWF) residues 1261-1478 (designated KB-AT-23-fus), and a second consisting of the bivalent control sdAb KB-hFX-11 also fused to the same VWF polypeptide (KB-hFX-11-fus). KB-hFX-11 does not bind to murine proteins, while VWF polypeptide is used for detection of both sdAbs. Following intravenous tail vein infusion (10 mg/kg) in wild-type C57BL/6 mice, recovery at 5 min was 93 ± 18% for KB-AT-23-fus and 47 ± 7% for KB-hFX-11-fus ($P = 0.012$). Both proteins were eliminated in a bi-exponential manner (Fig 3B). The half-lives of the initial portions (alpha-phase) of the decay curve were 0.3 h (95% CI 02–0.6 h) and 0.03 h (95% CI 0.02–0.05 h) for KB-AT-23-fus and KB-hFX-11-fus, respectively. The half-lives of the terminal portions (beta-phase) were 38 h (95% CI 21–178 h) and 0.7 h (95% CI 0.5–1.0 h) for KB-AT-23-fus and KB-hFX-11-fus, respectively. The notion that KB-AT-23-fus has a substantial longer half-life compared to the similar-sized control protein strongly suggests that KB-AT-23 associates with endogenous antithrombin of the mouse.

We next investigated the use of KB-AT-23 to reduce bleeding in a mouse model of severe hemophilia A (Bi *et al*, 1995). To do so, we

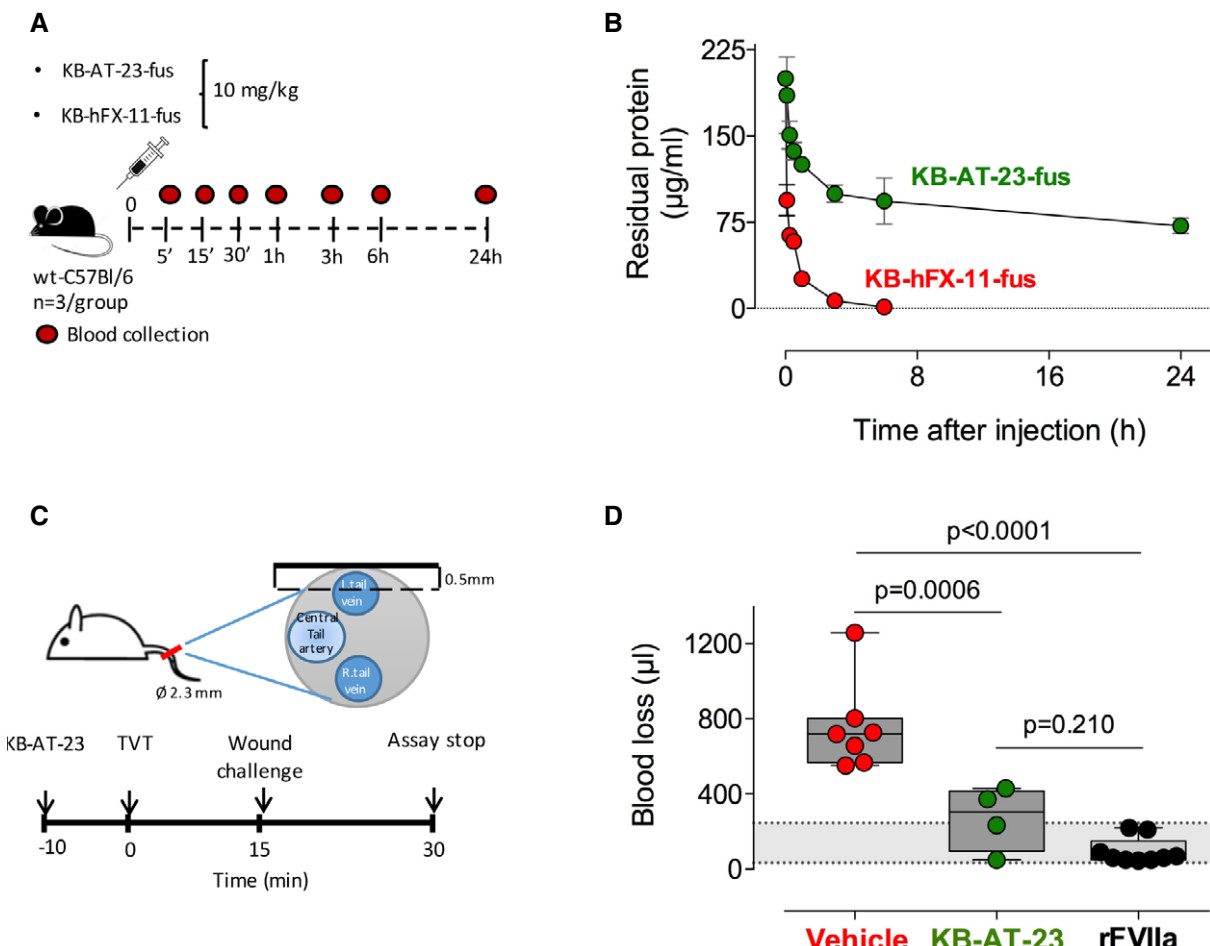

**Figure 3. Assessment of KB-AT-23 efficacy in hemophilia A mice delivered with the purified sdAb.**

A  Schematic representation of the clearance experiment.

B  Purified KB-AT-23-fus and KB-hFX-11-fus were given intravenously to wild-type C57B6 mice (10 mg/kg). At indicated time points, blood was collected and residual protein was measured. Plotted is residual protein versus time after injection. Green circles: KB-AT-23-fus; red circles: KB-hFX-11-fus. Data represent mean ± SD, with $n = 3$ for each data point.

C  Schematic representation of the tail vein transection (TVT) assay performed in $F8^{-/-}$ mice. Ø 2.3 mm: size of the diameter of the tail at the location of the TVT; 0.5 mm: depth of the cut performed on the mouse tail.

D  Graph reporting the volume of blood loss (µl) in treated mice ($n = 4$–9 per group) observed during 30 min post-TVT. Dosing was 10 mg/kg for KB-AT-23 and 1 mg/kg for FVIIa. The gray area represents the range of blood loss in FVIII-treated mice (33–245 µl), which is similar to that of wild-type C57BL/6 mice (49–308 µl; Johansen et al, 2016). Boxes represent 25–75% quartiles, the line indicates the median. Whiskers span min to max. Data were analyzed via 1-way ANOVA with Tukey's correction for multiple comparisons.

used a recently developed tail vein transection bleeding model (TVT) (Johansen *et al*, 2016), which involves the standardized transection of the lateral tail vein at a depth of 0.5 mm (Fig 3C). FVIII-deficient mice were given a single dose of KB-AT-23 (10 mg/kg), recombinant FVIIa (1 mg/kg), or control vehicle, and 10 min after injection, bleeding was assessed. Blood was collected over a period of 30 min, and blood loss was quantified as described (Muczynski *et al*, 2018). On average, blood loss was 755 ± 239 µl ($n = 7$) in vehicle-treated FVIII-deficient mice and 95 ± 69 µl ($n = 9$) in FVIIa-treated mice (Fig 3D). In contrast, mice receiving 10 mg/kg of KB-AT-23 showed reduced blood loss (272 ± 170 µl; $n = 4$; $P = 0.0006$ versus vehicle-treated mice and $P = 0.210$ versus FVIIa-treated mice; Fig 3D). These results indicate that KB-AT-23, administered as recombinant protein, efficiently reduces blood loss *in vivo* in the absence of FVIII.

## KB-AT-23 can be efficiently produced and secreted *in vitro* in hepatocyte cell lines

Based on the promising results obtained *in vivo* with KB-AT-23 administered as a recombinant protein, we next explored the possibility of constitutively expressing the protein in liver using an AAV vector-mediated gene transfer. In order to develop hepatotropic AAV vectors expressing secretable nanobodies, we first cloned the KB-AT-23 coding sequence carrying a N-terminal His-tag and a C-terminal human influenza hemagglutinin (HA) tag in a hepatocyte-specific expression cassette (Fig 4A). We then generated 5 variants with different N-terminal signal peptides, either derived from heavy chain of human immunoglobulins (Haryadi *et al*, 2015) or derived from a protein expressed from the liver and already described to efficiently

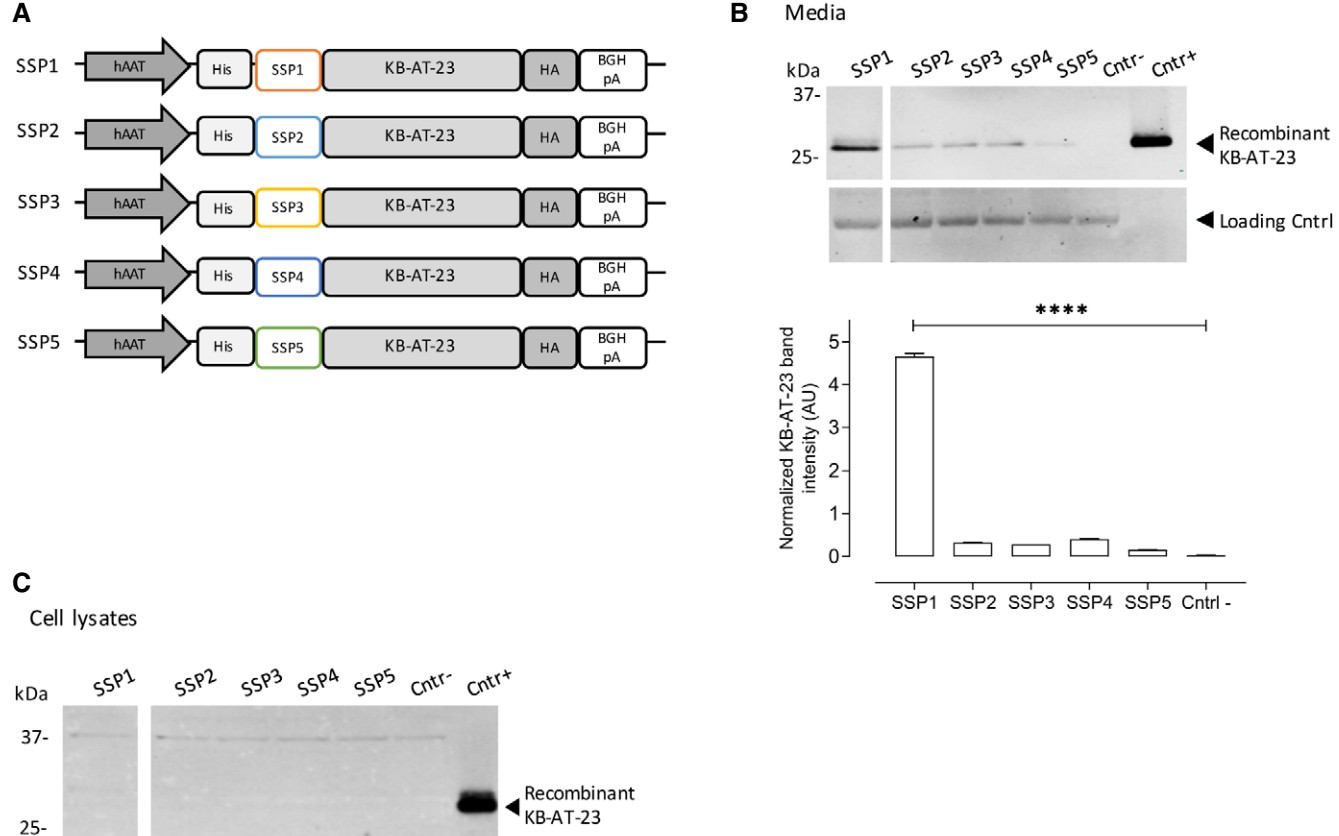

**Figure 4. Optimization of a liver-specific vector expressing a secretable KB-AT-23 sdAb variant.**

A   Schematic representation of the KB-AT-23 expression cassettes used to test the secretion signal peptides SSP1, SSP2, SSP3, SSP4, and SSP5. hAAT: human alpha-1 antitrypsin promoter; His: N-terminal 6×-Histidine tag; HA: C-terminal tag derived from the human influenza hemagglutinin; BGHpA: bovine growth hormone polyadenylation signal.

B   Western blot analysis of KB-AT-23 protein secretion in conditioned media 72 h post-transfection of HuH7 cells. In the lower panel, the Western blot quantification via densitometry analysis is depicted. Data are presented as mean ± SD and were analyzed via 1-way ANOVA with Dunnett's correction for multiple comparisons. ****$P < 0.001$.

C   HuH7 cells were collected 72 h post-transfection. Presented is the Western blot analysis of lysed cells for the presence of KB-AT-23. Cntr- refers to non-transfected cells, while Cntr+ refers to purified recombinant KB-AT-23.

Source data are available online for this figure.

**Table 4. List of secretion signal peptides screened for the KB-AT-23 expression cassette optimization.**

| Expression cassette | Short name | Signal Peptide Origin | Amino acid sequence |
|---|---|---|---|
| hAAT-SSP1-KB-AT-23 | SSP1 | Chymotrypsinogen B2 | MAFLWLLSCWALLGTTFG |
| hAAT-SSP2-KB-AT-23 | SSP2 | Human IgA1, IgG1, IgD heavy chain | MELGLSWIFLLAILKGVQC |
| hAAT-SSP3-KB-AT-23 | SSP3 | Human IgM, IgG2 heavy chain | MELGLRWVFLVAILEGVQC |
| hAAT-SSP4-KB-AT-23 | SSP4 | Human IgA1, IgE, IgG heavy chain | MDWTWRILFLVAAATGAHS |
| hAAT-SSP5-KB-AT-23 | SSP5 | Human IgG4 heavy chain | MEFGLSWVFLVALFRGVQC |

drive secretion of recombinant proteins into the bloodstream (Puzzo *et al*, 2017) (Fig 4A and Table 4). Human hepatoma HuH7 cells were transfected, and protein expression and secretion in conditioned media were assessed in samples collected 72 h later. Western blot analysis revealed a 30-kDa band corresponding to the secreted KB-AT-23 in all samples (Fig 4B), while in cell lysate, the protein was undetectable (Fig 4C). The highest amount of secreted KB-AT-23 was obtained with the SSP1 construct, as confirmed by the Western blot

quantification (Fig 4B). Based on these results, we identified an expression cassette optimally suited for liver expression of KB-AT-23.

**Long-term expression of KB-AT-23 in liver of HA mice is safe and effective in reducing bleeding**

We then assessed the safety and efficacy profile of KB-AT-23 stably secreted in liver from an AAV8 vector carrying the selected

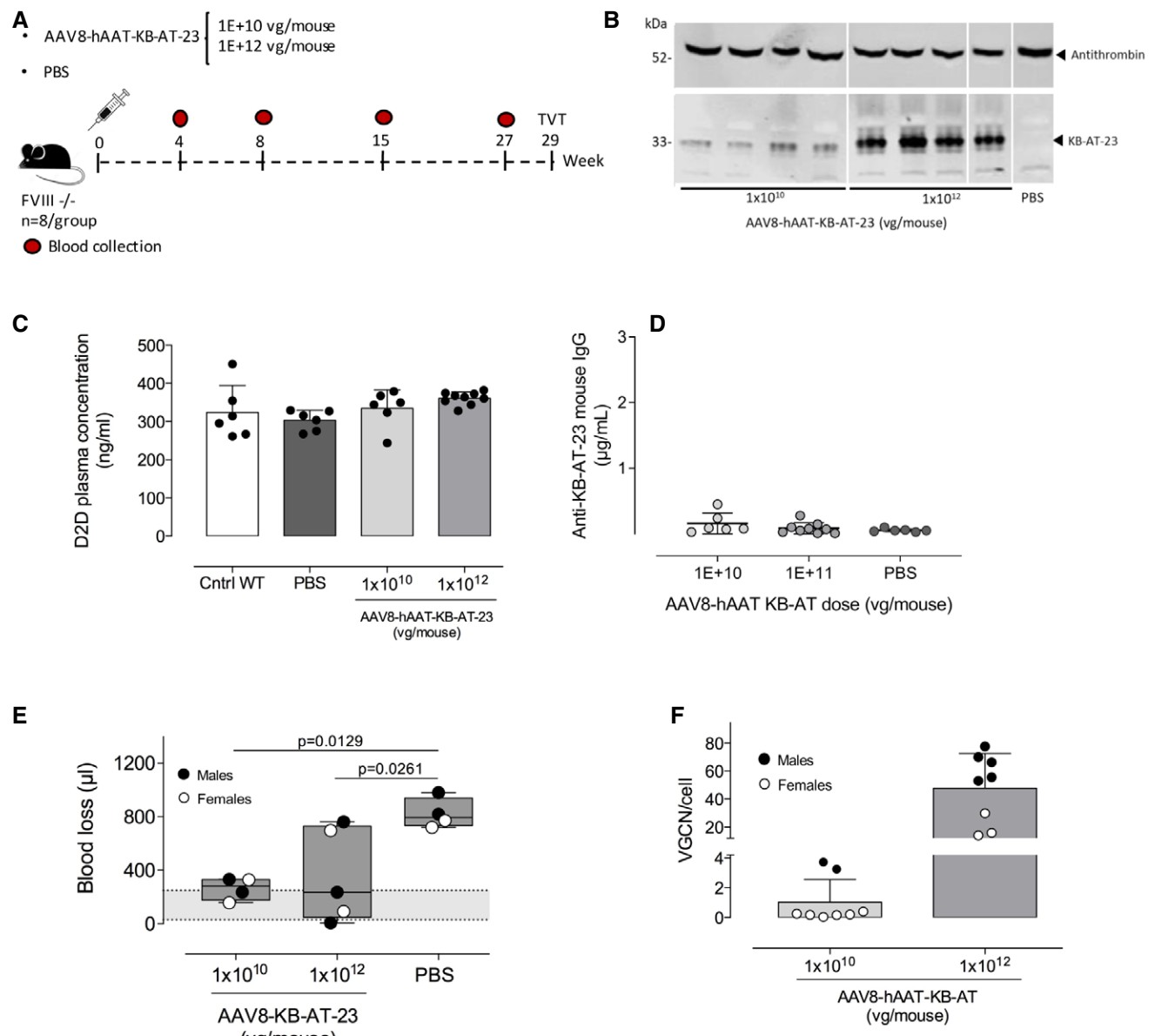

**Figure 5. Long-term safety and efficacy of KB-AT-23 delivered via AAV vectors in hemophilia A mice.**

A  Scheme representing the study design. FVIII-deficient mice (FVIII $^{-/-}$, $n = 6$–9/group) were administered with 2 doses ($1 \times 10^{10}$ or $1 \times 10^{12}$ vg/mouse) of AAV vector. Control mice were injected with PBS. Red symbols represent timing of blood collection.

B  Representative Western blot analyses of KB-AT-23 and antithrombin expression on plasma samples ($n = 4$) collected from animals 27 weeks post-AAV injection. kDa, molecular weight marker.

C  Measurement of D-dimers (D2D) plasma concentration 27 weeks post-vector administration. Data are presented as mean $\pm$ SD ($n = 6$–9/group).

D  Measurement of anti-KB-AT-23 mouse IgG in plasma samples collected from mice ($n = 6$–9/group) 8 weeks post-AAV administration. Data represent mean $\pm$ SD.

E  TVT test on males and females ($n = 4$–5 per group) performed 29 weeks post-vector administration. The volume of blood loss (μl) during 30 min of observation time is reported. The gray area represents the range of blood loss in FVIII-treated mice (33–245 μl), which is similar to that of wild-type C57BL/6 mice (49–308 μl; Johansen et al, 2016). Boxes represent 25–75% quartiles; the central line indicates the median. Whiskers span min to max. Data were analyzed via 1-way ANOVA with Dunnett's correction for multiple comparisons.

F  Vector genome copy number (VGCN) per cell in liver samples collected at sacrifice ($n = 8$/group). Data represent mean $\pm$ SD.

Source data are available online for this figure.

SSP1-KB-AT-23 expression cassette (AAV8-hAAT-KB-AT-23). Hemophilia A male and female animals received two doses of AAV8-hAAT-KB-AT-23 or PBS control intravenously ($1 \times 10^{10}$, and $1 \times 10^{12}$ vg/mouse, $n = 6$–9 per group) (Fig 5A) and were followed for 7 months. First, we assessed the circulating sdAb levels in plasma collected at different time points (Fig 5A). Western blot analyses revealed good correlation between the vector dose and the plasma levels of KB-AT-23 (Fig 5B, lower

panel). Since sdAbs exploit their inhibitory activity by binding antithrombin, we wanted to assess whether the circulating levels of KB-AT-23 had an impact on antithrombin levels. Western blot analysis of plasma antithrombin showed no significant differences between treated animals and PBS controls (Fig 5B, upper panel), indicating that the sdAb expression did not impact circulating antithrombin levels.

In order to evaluate potential adverse effects related to the sdAb expression, we measured the levels of D-dimers, a prothrombotic marker (Adam *et al*, 2009), in plasma samples collected at the end of the study, 7 months post-vector administration, and from normal wild-type mice as control. Following one-way ANOVA analysis with Tukey's multiple comparison test, no significant differences in D-dimer levels were observed between the groups (Fig 5C). Evaluation of anti-sdAb immune responses did not show development of IgG against the KB-AT-23 transgene (Fig 5D). At the end of the study (Fig 5A), a subset of animals from each treatment group ($n = 4$) was challenged in a TVT assay and blood loss was measured over 30 min. In both males and females, we observed a significant amelioration of the bleeding phenotype at both vector doses tested (Fig 5E), suggesting that the $1 \times 10^{10}$ vg/kg was already above the therapeutic threshold for correction of blood loss following TVT. After the sacrifice, vector genome copy number was measured in livers, showing a good correlation between the vector dose and the degree of liver transduction (Fig 5F). As previously reported (Davidoff *et al*, 2003), male animals displayed higher liver transduction levels than female animals (Fig 5F). These results demonstrate that long-term liver expression of KB-AT-23 via AAV gene transfer is safe and results in an amelioration of the bleeding diathesis in hemophilia A mice.

## Single-domain antibodies expressed via AAV vectors showed hemostatic activity in hemophilia B mice with or without antibodies directed against FIX

Based on the results obtained in hemophilic mice, we identified an intermediate vector dose of $2 \times 10^{11}$ vg/mouse to be tested in hemophilia B mice (Lin *et al*, 1997). We investigated the efficacy of correction of bleeding of KB-AT-23 in the presence or absence of antibodies directed against FIX. To do so, we pre-immunized mice ($n = 5$ per group) with human FIX protein (40 μg/mouse) formulated in complete Freund's adjuvant, or PBS control, injected subcutaneously (Fig 6A and B). Immunized

animals developed an anti-FIX immune response within 5 weeks following the FIX protein administration (Fig 6C). Six weeks post-immunization, mice were intravenously infused with the AAV8-hAAT-KB-AT-23 vector (Fig 6A). A group of animals was injected with PBS as control ($n = 5$) (Fig 6B). Western blot analysis of plasma samples at 8 weeks post-AAV injection showed that the KB-AT-23 transgene was correctly secreted in the bloodstream in the treated groups (Fig 6D). Consistently, we observed a significant reduction in AT activity following AAV vector administration in treated mice (~45% at 8 weeks post-injection, $P = 0.0001$, 2-way ANOVA) (Fig 6E). This resulted in a significant amelioration of the bleeding phenotype in both groups as assessed via a tail clip assay performed 9 weeks post-AAV vector administration (Fig 6F). Blood loss was reduced from $823 \pm 287$ μl (PBS-injected animals) to $328 \pm 153$ μl (mice injected with FIX+AAV) and $307 \pm 103$ μl (mice injected with PBS+AAV) (Fig 6F). Noticeably, KB-AT-23-treated mice displayed a blood loss comparable to that of the wild-type mice, which was $299 \pm 73$ μl and an AAV-FIX-treated group ($184 \pm 92$ μl) added as experimental controls (Fig 6F). No anti-sdAb antibody responses were detected in AAV vector-treated mice (Fig 6G). Together, these findings demonstrated that liver expression of KB-AT-23 can correct bleeding in hemophilia B mice regardless of the presence of anti-FIX antibodies.

## The bi-paratopic KB-AT-23 does not elicit anti-sdAb immune response in wild-type mice

In order to assess the potential immunogenicity of KB-AT-23 expressed *in vivo*, we produced AAV8 vectors expressing the bivalent KB-AT-23 or a trivalent KB-AT-113 nanobody under the control of the strong constitutive CMV early enhancer/chicken beta-actin (CAG) promoter (AAV8-CAG-KB-AT-23 and AAV8-CAG-KB-AT-113) (Fig 7A). Wild-type mice received either an intravenous infusion of AAV8-hAAT-KB-AT-23 ($1 \times 10^{10}$ vg/mouse, $n = 8$) or an intramuscular injection of the immunogenic vectors AAV8-CAG-KB-AT-23 and AAV8-CAG-KB-AT-113 ($1 \times 10^{10}$ vg/mouse, $n = 8$ per group) (Fig 7A and B). Protein expression analyses conducted a month post-AAV injection revealed the presence of circulating KB-AT-23 and KB-AT-113 sdAbs in the treated groups (Fig 7C). We then evaluated the immune response against the circulating sdAbs by performing an ELISA assay in which we used the monovalent KB-AT-01, KB-AT-02, or KB-AT-03 sdAb as capture antibodies

**Figure 6. Assessment of KB-AT-23 efficacy in hemophilia B mice with or without inhibitors.**

A  Scheme representing the study design. FIX-deficient mice (FIX$^{-/-}$, $n = 5$/group) were administered with factor IX (FIX, 40 μg/mouse) in the pre-immunization phase and with AAV vectors ($2 \times 10^{11}$ vg/mouse) in the treatment phase. Control mice were injected with PBS. Red symbols represent timing of blood collection.

B  Table reporting the different mice groups of the study.

C  Measurement of anti-hFIX mouse IgG in plasma samples collected from mice injected with FIX or PBS ($n = 5$/group) at different time points. Data represent mean ± SEM.

D  Representative Western blot analyses on plasma samples ($n = 4$) collected from animals 8 weeks post-AAV injection.

E  Antithrombin activity in plasma samples collected from mice at different time points. Data are presented as mean ± SD ($n = 3$–4 per datapoint). Data were analyzed in a 2-way ANOVA with Tukey's correction for multiple comparisons. ***$P < 0.01$ ****$P < 0.001$ (2-way ANOVA with Tukey's multiple comparison test).

F  Graph reporting the volume of blood loss (μl) over 30 min following the tail clip. WT: wild-type littermates of F9$^{-/-}$ mice. The gray area represents the range of blood loss of F9$^{-/-}$ mice that received AAV vectors expressing human FIX. Data are presented as mean ± SD ($n = 3$–4/group) and were analyzed in a 1-way ANOVA with Tukey's correction for multiple comparisons. **$P < 0.01$. Boxes represent 25–75% quartiles, with the line indicating the median. Whiskers span min to max.

G  Measurement of anti-KB-AT-23 mouse IgG in plasma samples collected from mice ($n = 5$/group) 8 weeks post-AAV administration. Data represent mean ± SD.

Source data are available online for this figure.

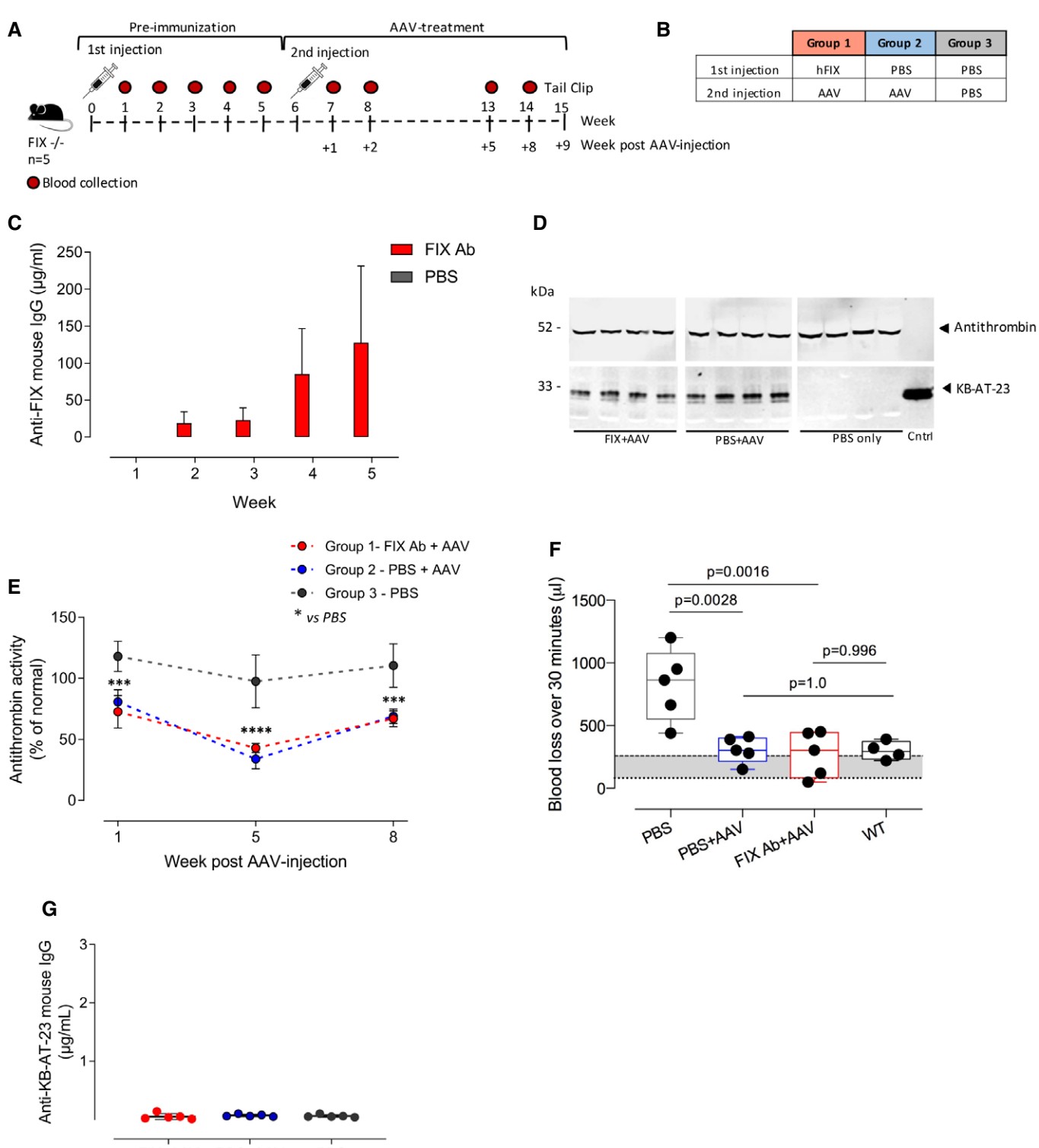

**Figure 6.**

(Fig 7D–F). These were the same sdAbs combined for the generation of the multivalent KB-AT-23 and KB-AT-113. Interestingly, we detected a consistent level of mouse IgG cross-reacting with the monovalent KB-AT-01 and KB-AT-02 in mice expressing the trivalent KB-AT-113 (Fig 7D and E). Conversely, we detected no or very low cross-reactivity against KB-AT-01 and KB-AT-02 in animals that received AAV vectors expressing KB-AT-23 under the control of the hAAT or CAG promoters (Fig 7D and E). None of the injected animals showed a significant humoral response directed against KB-AT-03 (Fig 7F). When we checked for the cross-reactivity of the mouse antibodies against the bivalent KB-AT-23 (Fig 7G) and the trivalent KB-AT-113 (Fig 7H), we observed an immune response

only in mice treated with the AAV8-CAG-KB-AT-113 (Fig 7G and H). On the other hand, mice expressing the bivalent KB-AT-23 either from the hAAT or from the CAG promoter did not show any

immune response (Fig 7G and H). Together, these results suggest that KB-AT-23 expressed via AAV gene transfer is poorly immunogenic.

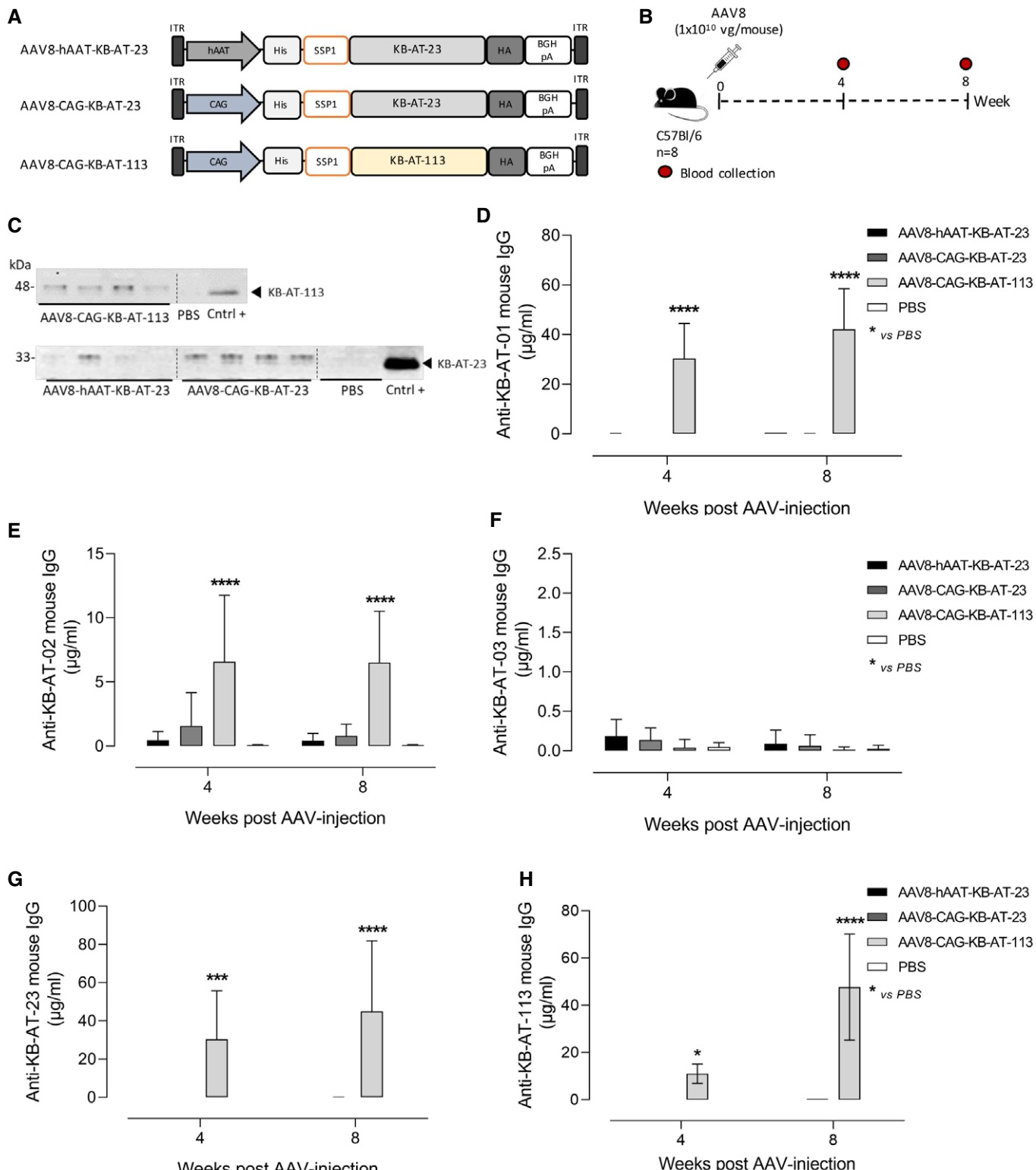

Figure 7.

**Figure 7. Assessment of sdAbs potential immunogenicity in wt mice.**

A Schematic representation of the AAV8 vectors used to express KB-AT-23 and KB-AT-113 variants. ITR: inverted terminal repeats for AAV packaging.

B Scheme representing the study design. Vectors were injected in C57Bl/6 mice (*n* = 8) at a dose of 1E+10 vg/mouse. Red symbols represent timing of blood collection.

C Representative Western blot analyses on plasma samples (*n* = 4) collected from animals 4 weeks post-AAV injection. Cntrl +: positive controls represented by the loading of the purified KB-AT-23 or KB-AT-113 proteins.

D–H Measurement of anti-sdAb mouse IgG in plasma samples collected at days 28 and 57 post-AAV injection. The ELISA plate was coated with the purified KB-AT-01 (D), KB-AT-02 (E), KB-AT-03 (F), KB-AT-23 (G), or KB-AT-113 (H) sdAbs. Data are reported as mean ± SD (*n* = 8/group). Data were analyzed in a 2-way ANOVA with Dunnett's correction of multiple comparisons. *$P < 0.05$, ***$P < 0.01$ ****$P < 0.001$.

Source data are available online for this figure.

## Discussion

In recent years, the landscape of treatments for hemophilia has changed dramatically. Long-acting clotting factors and non-factor replacement therapies have improved the management of bleeds and reduced the burden of treatment. Among the novel therapies for hemophilia, gene therapy trials for both hemophilia A and hemophilia B have also yielded impressive results, with some AAV vector-based gene therapies now in clinical trials (Nathwani *et al*, 2014; George *et al*, 2017; Rangarajan *et al*, 2017). Additional drug candidates to treat hemophilia are also in development, highlighting the need to address all complexities of the disease (Nogami & Shima, 2019; Pelland-Marcotte & Carcao, 2019). The ability of compensating the bleeding diathesis of hemophilia by acting on the feedback loop of the clotting cascade has been previously explored, for example, by crossing hemophilic mice with mice carrying the prothrombotic gene variant factor V Leiden (Schlachterman *et al*, 2005). Similarly, Shetty and coworkers reported the case of two individuals with markedly mild bleeding phenotype resulting from the combination of homozygous mutations associated with severe hemophilia and heterozygous for mutations resulting in antithrombin deficiency (Shetty *et al*, 2007). Here, we developed a novel potential therapeutic approach for both hemophilia A and hemophilia B based on the inhibition of antithrombin, which can restore clotting even in the presence of antibodies directed against a coagulation factor. Our strategy is based on single-domain antibodies binding to antithrombin that were derived from a library of llama single-domain antibodies. The combination of two sdAbs from the library, produced as a recombinant protein, was used to restore clotting both *in vitro* and *in vivo*. Single-domain antibodies, or nanobodies, have emerged in recent years as a novel class of biologics, owing to several attractive features (Aymé *et al*, 2017; Li *et al*, 2018), including ease of engineering to generate multiple binding domains and the potential for lower costs of manufacturing compared to monoclonal antibodies (Hassanzadeh-Ghassabeh *et al*, 2013; Steeland *et al*, 2016). Potential limitations include the variable half-life of the sdAbs, which are generally eliminated very rapidly from the blood due to their small size, particularly when not bound to a circulating protein (Fig 3B). To address this limitation, various protein engineering strategies are commonly used to extend the half-life of the sdAbs (Hoefman *et al*, 2015). Furthermore, while derived from llama and therefore of non-human origin, immunogenicity of sdAbs is relatively weak because of the high degree of identity with the human type 3 VH domain (VH$_3$) (Vu *et al*, 1997). Here, we showed how simple protein engineering can be applied to single-domain antibodies, by combining monomers targeting different domains of a functional protein to achieve enhanced reduction of its activity. A similar strategy to the one presented here, based on

the use of fitusiran, an siRNA targeting the antithrombin mRNA, is currently in late-stage clinical development. Clinical data indicate that a reduction in antithrombin levels by 70–90% can be achieved depending on the dose of siRNA administered (Pasi *et al*, 2017). Accordingly, an increase in thrombin generation was observed when tested in global thrombin generation assays, resulting in a significant reduction in the annualized bleeding rate in hemophilic patients (Pasi *et al*, 2017). While promising, the use of fitusiran has several potential drawbacks associated with the long delay (2–4 weeks) needed to achieve the desired reduction in antithrombin levels (Pasi *et al*, 2017), which makes the approach unsuitable to address acute bleeds. Additionally, because of the relatively low turnover of the siRNA delivered, the rate of recovery in antithrombin levels after discontinuation of the therapy is slow, reaching 10–15% per month (Pasi *et al*, 2017; Machin & Ragni, 2018), which may pose concerns over the ability to acutely discontinue the treatment, if needed. The use of sdAbs directed against antithrombin could potentially address these concerns. Indeed, while fitusiran prevents antithrombin synthesis by targeting and degrading antithrombin mRNA in the liver (Sehgal *et al*, 2015), sdAbs act as neutralizing agents for antithrombin, thus explaining the fast restoration of thrombin generation *in vitro* shown here. Based on kinetic progress curves in which KB-AT-23 neutralizes antithrombin-mediated inhibition of thrombin and FXa, it appears that KB-AT-23 behaves as a tight binding, competitive inhibitor. This suggests that KB-AT-23 interferes with complex formation between antithrombin and the enzymes. Future work will be aimed at elucidating the mechanism of action and the specific epitopes in more detail. It is worth noting that the duration of gene silencing by fitusiran is strongly dependent on the siRNA intracellular turnover (Bartlett & Davis, 2006), while KB-AT-23 activity depends entirely on its binding to circulating antithrombin. Free KB-AT-23 is rapidly eliminated, while antithrombin-bound KB-AT-23 is removed from the circulation in an antithrombin-dependent manner. This allows for a rapid reversal of the treatment if needed, for instance in the event of thrombosis (Dargaud *et al*, 2004). In other emergency situations, the infusion of antithrombin concentrates could be used to directly neutralize the action of KB-AT-23 and normalize levels of antithrombin. Indeed, the addition of antithrombin concentrate to sdAb-containing plasma reversed thrombin generation in these samples (Table 3).

In our *in vitro* thrombin generation experiments, using a molar excess of sdAb over antithrombin, we noticed that the ETP was increased 1.5- to 2-fold (Table 2), potentially raising questions on whether this approach would pose an increased risk of thrombosis. However, it should be pointed out that in the plasma-based *in vitro* thrombin generation assay, the anticoagulant pathways are underrepresented: Only 10% of the tissue factor pathway inhibitor

molecules is present, while cellular thrombomodulin (needed to activate the activated protein C pathway) and protease nexin-1 (a strong inhibitor of thrombin released from platelets) are both absent in these assays. It stands to reason therefore to assume that the *in vitro* thrombin generation assay in FIX- or FVIII-deficient plasma in the presence of an antithrombin inhibitor can result in artificially exaggerated thrombin generation. Indeed, the presence of KB-AT-23 results in near-normalization of the bleeding tendency, while D-dimer levels (a marker for thrombosis) were not increased even after prolonged exposure to KB-AT-23. Based on these considerations, sdAbs could become a useful tool to improve the prophylactic treatment of hemophilia.

With an eye on the rapidly evolving landscape of gene therapy for hemophilia, which includes both gene replacement approaches (Nathwani *et al*, 2011, 2014; George *et al*, 2017; Rangarajan *et al*, 2017; Miesbach *et al*, 2018) and preclinical approaches based on alternative pathways (Schuettrumpf *et al*, 2005; Quade-Lyssy *et al*, 2014), here we also exploited AAV vector-mediated gene transfer as a modality to express our anti-antithrombin sdAb *in vivo* in hemophilic mice. To this aim, we engineered an AAV vector to drive liver secretion of the sdAb into the bloodstream. Potential advantages of this approach over the conventional protein-based therapy are the possibility of achieving a long-term steady-state level of expression, which would avoid peak and trough associated with protein-based therapy thus resulting in improved efficacy. From a preclinical evaluation point of view, AAV-mediated gene transfer allowed us to evaluate both the short- and long-term safety and efficacy profile of our investigational therapy. Results shown here demonstrate that liver expression of our anti-antithrombin sdAb KB-AT-23 can control bleeding in both hemophilia A and B mice, in a model of a major vascular injury. As expected, bleeding was also controlled in hemophilia B mice that were previously immunized against FIX, suggesting that control of bleeding can be obtained also in the presence of inhibitors. Thus, these combined results provide proof of concept of the feasibility of this investigational therapeutic strategy for both hemophilia A and hemophilia B. Translating these findings to humans, particularly as a gene therapy, will required further safety and efficacy studies in small and large animal models of hemophilia (Sabatino *et al*, 2012), particularly aimed at defining the minimal effective vector dose that would result in correction of bleeding time. Additionally, the use of inducible promoters (Chaveroux *et al*, 2016) may be necessary to regulating the sdAb expression over time. Nevertheless, although preliminary, data in hemophilia A mice suggest that long-term (7-month) inhibition of antithrombin activity (~45%) via liver expression of a bivalent sdAb is safe, as it did not increase significant levels of D-dimers nor induce an immune response in the treated animals. Future preclinical studies with single-domain antibodies, along with data emerging from ongoing late-stage trials (ClinicalTrials.gov ID: #NCT03754790, #NCT03417102, #NCT03417245), will help further assessing the thrombogenic risk associated with the long-term inhibition of antithrombin. Taking advantage of the AAV vector system, we also carefully explored the immunogenicity profile of our therapeutic candidate KB-AT-23 *in vivo*. As it is well established that the liver expression of a protein is associated with induction of immune tolerance (Mingozzi *et al*, 2003; Puzzo *et al*, 2017), we expressed our sdAb under the control of a strong constitutive promoter known to drive immunogenicity (Colella *et al*, 2019) and expressed it via an

AAV vector. Even under these highly immunogenic conditions, we were not able to detect a significant response against KB-AT-23. In contrast, a trivalent sdAb used in parallel elicited strong anti-sdAb antibody formation. The fact that no immune response was detected against the bivalent KB-AT-23 under tolerogenic (liver-restricted expression) or immunogenic (constitutive expression) condition is a clear indication that the immunogenicity profile of this investigational therapeutic candidate is low. One possible explanation of these results is that a small-size protein (33 kDa for KB-AT-23) that is secreted (Perrin *et al*, 2016; Puzzo *et al*, 2017) has a low immunogenicity profile, while a larger trivalent sdAb would carry a higher risk of triggering humoral responses, particularly in the setting of delivery as recombinant protein. In summary, here we show for the first time that it is possible to restore clotting in mouse models of hemophilia A and hemophilia B using engineered single-domain antibodies directed against antithrombin, delivered as recombinant protein or as gene therapy. Liver expression of a bivalent sdAb restored clotting in hemophilia A mice and in hemophilia B mice, even in the presence of anti-FIX antibodies. These results provide proof of concept supporting the feasibility of the approach and warranting future efforts to translate these results to humans.

# Materials and Methods

### Immunization protocol

Immunization of a single llama (*L. glama*) was outsourced to the Centre de Recherche en Cancérologie (Université Aix-Marseille, Marseille, France). The animal was given 0.4 mg of human antithrombin with Freund's incomplete adjuvant at weekly intervals. Blood was collected at 4 weeks after the first injection for the isolation of peripheral blood lymphocytes.

### Construction of the sdAb library from lymphocytes

Total mRNA from B lymphocytes was used for the construction of an sdAb library as described (Behar *et al*, 2008; Aymé *et al*, 2017). Briefly, total mRNA was used for the synthesis of cDNA via reverse transcriptase (Roche, Meylan, France) using the CH2′ primer (5′-GGTACGTGCTGTTGAACTGTTCC-3′) as described previously (Arbabi Ghahroudi *et al*, 1997). sdAb-coding DNA fragments were obtained from the cDNA by a nested-PCR, and fragments were subsequently cloned into the pHEN6 phagemid vector (Hoogenboom *et al*, 1991). The ligated material was used to transform electrocompetent *E. coli* TG1 cells (Thermo Fischer Scientific, Villebon-sur-Yvette, France), allowing the creation of a library containing $1.1 \times 10^8$ clones.

### Enrichment of phages that express antithrombin-specific sdAb

To capture phages expressing antithrombin-specific sdAb, phage particles (500 μl) were incubated with Dynabeads M-450 epoxy beads (200 μl; Thermo Fischer Scientific) coated with purified human antithrombin (1 mg/ml beads) for 1 h at room temperature in PBS/3% BSA. As a control, phage particles were incubated with BSA-coated beads. Beads were washed nine times using PBS/0.1% Tween-20 (PBS/T) and twice with PBS. Captured phages were eluted via incubation with 0.5 mg/ml trypsin (1 ml volume). Eluted

phages (10 µl) were serially diluted to infect *E. coli* TG1 cells and plated to evaluate the antigen-specific enrichment.

### Selection of antithrombin-specific sdAb

To isolate true antithrombin-specific sdAb, 282 TG1 clones infected with phages eluted from antithrombin-coated beads were grown overnight in 0.5 ml TB medium and sdAb expression was induced via IPTG (1 mM). Periplasmic extracts were prepared as described (Habib *et al*, 2013) and tested for binding to antithrombin using antithrombin- and BSA-coated microtiter wells. Bound sdAb was probed using peroxidase-coupled polyclonal anti-HIS antibodies (Dilution 1/2,000; Abcam, Paris France) and detected via hydrolysis of 3,3′,5,5′-tetramethylbenzidine (TMB).

### sdAb subcloning, expression, and purification

Selected sdAb cDNA sequences were cloned into the pET28 plasmid allowing intracytoplasmic bacterial expression. Plasmids encode sdAb with an N-terminal His-tag and a C-terminal haemagglutinin (HA)-tag to facilitate purification and detection, and were used for expression of sdAbs in *E. coli*-Shuffle-C3029H cells (New England Biolabs; Evry, France).

Bacteria were grown in 330 ml of LB medium at 30°C until the culture had reached an optical density between 0.4 and 0.6. Protein expression was then induced at 20°C via the addition of isopropyl-thiogalactoside (0.1 mM final concentration). Twenty-four hours after induction of protein expression, cells were collected, and soluble proteins were released from the cells via sonication. His-tagged sdAbs were purified via $Co^{2+}$-affinity chromatography using PBS/0.3 M Nacl/0.5 M imidazole as elution reagent. Minor contaminants were removed from the preparations via size-exclusion chromatography using 20 mM HEPES (pH 7.4)/0.1 M NaCl as equilibrium buffer. Purified sdAbs displayed > 95% homogeneity as assessed via SDS–PAGE and Coomassie staining.

### Immunosorbent assays

Binding of antithrombin to immobilized purified monovalent sdAbs or derivatives thereof was performed as follows. sdAbs were immobilized (5 µg/ml) in a volume of 50 µl in half-well microtiter plates for 16 h at 4°C. After washing, wells were incubated with purified human or murine antithrombin (0–5 µM) or with plasma (diluted fourfold) of different species in Tris-buffered saline (pH 7.6) supplemented with 5% skimmed milk for 2 h at 37°C. Bound antithrombin was probed using peroxidase-labeled polyclonal rabbit anti-antithrombin antibodies (Dilution 1/100; Stago BNL, Leiden, The Netherlands) and detected by measuring peroxidase-mediated hydrolysis of 3,3′,5,5′-tetramethylbenzidine (TMB).

### Antithrombin activity assay

Purified antithrombin (2 nM) was incubated in the absence or presence of sdAbs (100 nM) for 15 min in TBSC buffer (Tris-buffered saline (pH 7.4), supplemented with 50 mM $CaCl_2$, 0.1% protease-free BSA, 0.1% PEG8000) at 37°C. This mixture was subsequently added to thrombin (0.5 nM) or FXa (0.5 nM) in the absence or presence of heparin. Residual activity of thrombin or FXa was measured using substrates S-2238 and S-2765, respectively.

### Thrombin generation assay

Thrombin generation in platelet-poor plasma was measured in a microtiter-plate fluorometer (Fluoroskan Ascent, Thermo Labsystems, Helsinki, Finland) according to the method described by Hemker *et al* (2002) with phospholipid and tissue factor concentrations being 4 µM and 1 pM, respectively. Endogenous thrombin potential (ETP), *i.e.,* area under the curve, thrombin peak, and lag time for thrombin detection, was determined using dedicated software (Thrombinoscope BV, Maastricht, The Netherlands). Immuno-depleted FVIII plasma (Diagnostica Stago, Asnières, France) was supplemented with recombinant purified FVIII (rFVIII, 0–100% U/ml Kogenate FS, Bayer HealthCare, Puteaux, France) or purified KB-AT-23. For experiments using murine FVIII-deficient plasma, 0.75 pM tissue factor was used instead of 1 pM. Polyclonal sheep anti-FVIII antibodies (titer 2,500 BU/ml) were used as inhibitory antibodies and were applied at a final titer of 8 BU/ml. Aclotine (LFB Biomedicaments, Les Ulis, France) was used as antithrombin concentrate at a final concentration of 2 µM.

### Generation of sdAb expression cassettes and AAV vectors

The sdAb transgene expression cassettes used in this study contained a bivalent sdAb named KB-AT-23 composed of two monovalent antibodies (KB-AT-02 and KB-AT-03) and a trivalent sdAb named KB-AT-113 composed of 2 monovalents KB-AT-01 and KB-AT-03. The monovalent sdAbs were linked together via a triple Gly3Ser-linker. They were fused together with an N-terminal poly-histidine tag (His-tag) and a C-terminal human hemagglutinin tag (HA tag). The sdAb transgenes were cloned in *cis*-plasmids for AAV vector production under the transcriptional control of the hepatocyte-specific hAAT promoter composed by the apolipoprotein E enhancer (ApoE) and the human alpha-1 antitrypsin promoter (abbreviated as hAAT) (Miao *et al*, 2003; Ronzitti *et al*, 2016) or the ubiquitous or the CMV enhancer/chicken beta-actin promoter (CAG) promoter. All DNA sequences used in the study were synthetized either by GeneCust or by Thermo Fisher Scientific. The expression cassettes contained an intron between the promoter and the transgene start codon, a BGH polyA signal after the transgene stop codon and were flanked by the ITRs of AAV2. AAV vectors were prepared as previously described (Ayuso *et al*, 2010). Briefly, genome-containing vectors were produced in roller bottles following a triple transfection protocol with cesium chloride gradient purification. Titers of AAV vector stocks were determined using real-time qPCR performed in ABI PRISM 7900 HT Sequence Detector using Absolute ROX mix (TaqMan, Thermo Fisher Scientific, Waltham, MA) and SDS–PAGE, followed by SYPRO Ruby protein gel stain and band densitometry. The AAV serotype used in the *in vivo* experiments was AAV8.

### *In vitro* studies

HuH7 cells were maintained under 37°C, 5% $CO_2$ condition in Dulbecco's modified Eagle's Medium (DMEM) supplemented with 10% FBS, 2 mM GlutaMAX (Thermo Fisher Scientific, Waltham, MA). Cells were seeded into 12-well plates ($2 \times 10^5$ cells/well) and

transfected with plasmids encoding for KB-AT-23 (1 μg/well) using Lipofectamine 2000 (Thermo Fisher Scientific) accordingly to the manufacturer's instructions. A plasmid encoding for EGFP under the control of the phosphoglycerate kinase (PGK) promoter (2 μg/well) was transfected as control. Seventy-two hours after transfection, cells and conditioned media were harvested and analyzed for KB-AT-23 expression via Western blot analyses.

### In vivo studies

All animal experiments were performed in strict accordance with good animal practices following French and European legislation on animal care and experimentation (2010/63/EU).

Immunization of a single llama (L. Glama) was performed by an outside contractor (Centre de Recherche en Cancérologie, Marseille, France) as a fee-for-service. Llama management, inoculation, and sample collection were conducted by trained personnel under the supervision of a veterinarian, in accordance with protocols approved by the local ethical committee of animal welfare of the Centre National de la Recherche Scientifique. Mouse experiments at Inserm U1176 were approved by the local ethical committee CEEA26 (protocol APAFIS#4400-2016021716431023v5), and experiments at Genethon were approved by the local ethical committee CERFE-Genopole (protocol n.C 91-228-107). Wild-type C57BL/6 mice were purchased at Charles River. F9 knock-out mice ($F9^{-/-}$) were purchased from The Jackson Laboratory (B6;129P2-F9tm1Dws/J, Stock No 004303) (Lin *et al*, 1997). F8-deficient mice ($F8^{-/-}$ mice) have been backcrossed (> 10 times) on a C57Bl/6 background (Bi *et al*, 1995). In the experiments with $F8^{-/-}$ mice, male/females mice aged 6–8 weeks were used. In the experiments with $F9^{-/-}$ mice, male mice aged 6–8 weeks were used. Mice received AAV8 vectors (100 μl) via a retro-orbital injection. In the study with anti-hFIX antibodies, mice received 100 μl of FIX in complete Freund's adjuvant via subcutaneous injection. Blood samples were collected from the retro-orbital plexus in 3.8% citrate-coated capillary tubes (Hirschmann Laborgeräte, Germany). At euthanasia, tissues were collected and snap-frozen for additional studies.

### Western blot analyses

HuH7 cell lysates were prepared using 10 mM PBS (pH 7.4) containing 1% of Triton X-100 and protease inhibitors (Roche Diagnosis). Western blot on mouse plasma was performed on samples diluted 1:4 in distilled water. Protein concentration was determined using the BCA Protein Assay (Thermo Fisher Scientific). SDS–page electrophoresis was performed in a 4–15% gradient polyacrylamide gel. After transfer, the membrane was blocked with Odyssey buffer (Li-Cor Biosciences) and incubated with an anti-HA tag antibody (Dilution 1/1,000; rabbit polyclonal PRB-101C, BioLegend) or anti-SERPINC1 (Dilution 1/1,000; rat monoclonal MAB1287, R&D System). The membrane was washed and incubated with the appropriate secondary antibody (Li-Cor Biosciences), and visualized by Odyssey Imaging System (Li-Cor Biosciences). For Western blot quantification, we used the Image Studio Lite Ver 4.0 software. The KB-AT-23 protein bands were normalized using an aspecific band detected by the anti-HA tag antibody in mouse plasma and used as loading control.

### The paper explained

#### Problem

Novel therapies for hemophilia, including non-factor replacement and *in vivo* gene therapy, are showing promising results in the clinic. However, challenges remain, including addressing patients with a history of factor VIII (FVIII) or IX (FIX) inhibitor development.

#### Results

We have developed a novel therapeutic approach for hemophilia, based on llama-derived single-domain antibody fragments (sdAbs). We engineered a bi-paratopic sdAb able to efficiently neutralize the anticoagulant activity of antithrombin *in vitro*, resulting in restoration of the thrombin generation potential in FVIII-deficient plasma, and correction of bleeding in a mouse model of acute bleeding injury. We showed that the long-term AAV vector-mediated expression of the bi-paratopic sdAb in hemophilia A and hemophilia B mice was safe and poorly immunogenic. Importantly, it also resulted in sustained correction of the bleeding phenotype, even in the presence of inhibitory antibodies to the therapeutic clotting factor.

#### Impact

We demonstrated that engineered single-domain antibodies directed against antithrombin have the potential of re-balancing hemostasis in a FVIII/FIX-independent manner, both when delivered as a protein or via liver-directed gene therapy.

### Measurement of anti-sdAb mouse antibodies

MaxiSorp 96-well plates (Thermo Fisher Scientific) were coated with 2.5 μg/ml of purified KB-AT-01, KB-AT-02, KB-AT-03, KB-AT-23, or KB-AT-113 proteins. IgG standard curves were made by serial 1/2 dilutions of commercial mouse recombinant IgG (Sigma-Aldrich) which were coated directly onto the wells in duplicate. Plasma samples appropriately diluted in PBS-0.1%-BSA, Tween 0.05% were analyzed in duplicate. An HRP-conjugated anti-mouse IgG antibody, Fc specific (Dilution 1/40,000; SIGMA, Ref. A9309), was used as secondary antibody.

### Clearance experiments in C57BL/6 wild-type mice

Proteins were generated in which human von Willebrand factor (VWF) residues 1,261–1,478 (with residue K1332 mutated to A) were fused to the C-terminal part of KB-AT-23. The K1332A mutation was included to ensure that the VWF polypeptide would not interact murine platelets. The resulting protein was designated as KB-AT-23-fus. As a control protein, we used KB-hFX-11-fus, in which the same VWF polypeptide was fused to the C-terminal end of a control nano-body (KB-hFX-11), that does not recognize murine proteins. Purified KB-AT-23-fus and KB-hFX-11-fus were given intravenously (10 mg/kg) to wild-type C57Bl/6 mice. At different time points after injection (5, 15, 30 min, 1, 3, 6 and 24 h), blood samples were obtained via retro-orbital puncture from isoflurane-anesthetized mice and plasma was prepared by centrifugation (1,500 *g* for 20 min at 22°C). Residual plasma concentrations were measured using an in-house ELISA that specifically measures the human VWF polypeptide, employing murine monoclonal antibody mAb712 (10 μg/ml) as capturing antibody and peroxidase-labeled murine monoclonal antibody mAb724 (Dilution 1/1,000) as probing antibody.

**Functional assessments in hemophilic mice**

For the TVT/tail clip procedures, mice were anesthetized with a mixture of ketamine and xylazine (100 and 16 mg/kg, respectively) injected intraperitoneally and a precise cut of the tip of the tail was performed as described (Liu, 2012; Johansen *et al*, 2016). At the end of the assay (30 min of observation time), animals were sacrificed by cervical dislocation. Mean values of the blood loss volume (μl) were reported.

**Vector genome copy number**

DNA from tissues was extracted after whole-organ homogenization using the QIAgen Blood Core Kit B precipitation method. Vector genome copy number was determined using a qPCR assay using 100 ng of DNA. The hAAT promoter-specific primers and the probe (forward primer 5′-GGCGGGCGACTCAGATC-3′, reverse primer 5′-GGGAGGCTGCTGGTGAA TATT-3′, probe (FAM) 5′-AGCCCCTG TTTGCTCCTCCGATAACTG-3′ (TAMRA)) were synthesized by Thermo Scientific (Waltham, MA, USA). Mouse titin was used as a normalizing gene (forward primer 5′-AAAACGAGCAGTGACGT GAGC-3′, reverse primer 5′-TTCAGTCATGCTGCTAGCGC-3′, probe (VIC) 5′-TGCACGGAAGCGTCTCGTCT CAGTC-3′ (TAMRA) synthesized by Thermo Scientific (Waltham, MA, USA)). Each sample was tested in triplicate.

**Statistical analysis**

All the data showed in the present manuscript are reported as mean ± standard deviation (SD). The number of sampled units, n, upon which we reported statistic, is the single mouse for the *in vivo* experiments (one mouse is $n = 1$). Statistical analyses were conducted with GraphPad Prism 7 software (GraphPad Software). For all the data sets, data were analyzed by Student's *t*-test for two-group comparisons or parametric tests (one- and two-way ANOVA with Tukey's or Dunnett's post hoc correction for comparisons with more than two groups). $P < 0.05$ were considered significant. The statistical analysis performed for each data set is indicated in figure legends. For all figures, $*P < 0.05$, $**P < 0.01$, $***P < 0.001$, $****P < 0.0001$.

**Expanded View** for this article is available online.

## Acknowledgements

We thank the Centre de Recherche en Cancérologie (Marseille, France) for performing immunization of the llama. This work was supported by the E-RARE grant ANR-15-RAR3-0008 (Agence Nationale de la Recherche, SMART-Haemo-Care to PJL & FM), a grant from the Association Françaises des Hémophiles (to ODC), a proof-of-concept grant from Inserm Transfert (to ODC and PJL), and an ERC-2013-CoG Consolidator Grant (grant agreement 617432, MoMAAV, to FM).

## Author contributions

EB and GA performed most of the experimental activities and data analyses. AM, J-FO, CK, CC, and SV contributed to mouse studies set-up and biochemical analyses. SM and LW performed animal procedures and the harvesting of mouse samples. SC and FC produced the AAV vectors used in the studies. CVD and ODC contributed to the interpretation of results and provided critical insights into the significance of the work. EB, FM, and PJL directed the work and wrote the manuscript.

## Conflict of interest

FM is employee and holds equity of Spark Therapeutics. GA, CVD, ODC, and PJL are inventors on a patent application on sdAbs against antithrombin. CVD, ODC, and PJL are founders and owners of Laelaps Therapeutics.

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
