## [Review Process File · EMBO Molecular Medicine]

Single-domain antibodies targeting antithrombin reduce bleeding in hemophilic mice with or without inhibitors

Elena Barbon, Gabriel Ayme, Amel Mohamadi, Jean-François Ottavi, Charlotte Kawecki, Caterina Casari, Sebastien Verhenne, Solenne Marmier, Laetitia van Wittenberghe, Severine Charles, Fanny Collaud, Cecile V. Denis, Olivier D. Christophe, Federico Mingozzi & Peter Lenting

Review timeline:

Submission date:	14th Aug 2019
Editorial Decision:	23rd Sep 2019
Revision received:	28th Jan 2020
Editorial Decision:	6th Feb 2020
Revision received:	14th Feb 2020
Accepted:	18th Feb 2020

Editor: Lise Roth

Transaction Report:

1st Editorial Decision

23rd Sep 2019

Thank you for the submission of your manuscript to EMBO Molecular Medicine, and please accept my apologies for the delay in getting back to you, which is due to the fact that one referee needed more time to complete his/her report. We have now received feedback from the three reviewers who agreed to evaluate your manuscript. As you will see from the reports below, the referees acknowledge the interest of the study and are overall supporting publication of your work pending appropriate revisions.

Addressing the reviewers' concerns in full will be necessary for further considering the manuscript in our journal, and acceptance of the manuscript will entail a second round of review. EMBO Molecular Medicine encourages a single round of revision only and therefore, acceptance or rejection of the manuscript will depend on the completeness of your responses included in the next, final version of the manuscript. For this reason, and to save you from any frustrations in the end, I would strongly advise against returning an incomplete revision.

***** Reviewer's comments *****

Referee #1 (Remarks for Author):

Recent studies have shown that siRNA-mediated reduction in plasma levels of the anticoagulant antithrombin can successfully be used to restore normal haemostasis in patients with severe haemophilia. In this paper, Barbon et al report an entirely novel approach to treat patients with haemophilia using single-domain antibodies targeting antithrombin. They show for the first time that these nanobodies can correct thrombin generation in haemophilic plasma, and also reverse the bleeding phenotype in murine models of haemophilia A and B. Importantly they further demonstrate the efficacy of sdAbs to attenuate bleeding phenotype even in the presence of inhibitory antibodies to FIX. This observation is particularly important from the clinical perspective, given the huge morbidity and mortality that continue to be associated with development of high responding inhibitors in patients with severe haemophilia. All together, the paper is well written and follows a

logical progression. The data presented are convincing and support the conclusions reached. These novel data are clearly of direct translational relevance, and will undoubtedly be of major interest to a wider readership.

I have a few minor suggestions that may improve the manuscript:

1. The effect of the sdAbs in down-regulating AT activity is convincing. However, some additional insight into the biological mechanisms through which this effect is achieved would be interesting. For example, it appears that the sdAbs are affecting AT activity rather than causing reduced antigen levels via enhanced clearance. Can different AT activity assays (eg heparin co-factor activity vs progressive activity) provide any additional insights into the mode of action of the sdAbs? Are the effects similar for AT inhibition of its different substrates? Is there any data on the epitopes recognised by the sdAbs ?
2. What is the half-life / MRT of recombinant KB-AT-23? From a clinical perspective, might be interesting to know whether the inhibitory effect of sdAbs can be overcome by addition of exogenous antithrombin concentrate.
3. In the discussion, the authors comment upon their mechanism of targeting AT (either with recombinant sdAb or with AAV mediated gene transfer) vis a vis that involving siRNA. I think it would be interesting to further expand this discussion - for example in terms of comparing (i) half-lives / duration of effect (ii) reversal potential (iii) respective effects on plasma AT antigen versus activity.
4. As noted above, the data demonstrating that the sdAb can ameliorate murine bleeding in the presence of a FIX inhibitor are of direct clinical importance. It would be nice to supplement this with data showing that the sdAB also corrects thrombin generation in human haemophilia plasma in the presence of a high responding inhibitor.

Referee #2 (Comments on Novelty/Model System for Author):

They should test their approach in hemophilia A mice with inhibitors.

Referee #2 (Remarks for Author):

In this manuscript, Barbon and colleagues reported that using single domain antibody fragments (sdAbs) to inhibit the antithrombin pathway can restore hemostasis in hemophilic mice. They showed that antibody against antithrombin was able to restore thrombin generation in vitro and improved hemostasis in hemophilia A mice. Then they targeted sdAb expression to hepatocytes by AAV and demonstrated the efficacy in hemophilia A mice without inhibitors and hemophilia B mice with inhibitors. The idea is novel. There are some concerns in the manuscript that needs to be addressed.

Concerns

- They showed (Figure 5F) that animals in the high dose group did get much higher copy number of viral genome per cell as determined by quantitative real-time PCR; however, the phenotypic correction in the high-dose group appears to be not as good as the low-dose group (Figure 5E). How to explain that the animal with 10-fold greater transduction efficacy and better KB-AT-23 expression but blood loss was similar to PBS-treated controls? Is it because animals produced antibodies against viral capsid protein or transduced hepatocytes were attacked by immune cells? Did liver enzyme increase in transduced animals?
- It is unclear why the manuscript does not include studies in hemophilia A with inhibitors. The authors showed the efficacy of targeting bi-paratopic sdAb expression in hemophilia B mice with inhibitors. They should include studies from hemophilia A with inhibitors to conclude that expression of bi-paratopic sdAb can result in sustained correction of the bleeding phenotype in hemophilic mice with inhibitors.
- They should include data from the WT control group in Figure 3B to show whether infusion of KB-AT-23 was able to completely rescue the bleeding phenotype in hemophilia A mice.
- Fig 1B, C, &D, There is a discrepancy between the figure legend and figure shown in Fig B, C, and D. The Figures show bar graphs, but figure legend describes "histogram". What is the N number in those groups?
- Figure 2B-C, what is the N number in those studies?

- Figure 3A, in the schematic diagram, it is unclear the relation between Ø2.3 mm and the depth of 0.5 mm. Also, there is a discrepancy between Figure legend and actual animal number as figure showed 7 data points in the Vehicle group, but the author described 4-6 in the figure legend.
- In Figure 4B, there is a discrepancy in the figure legend and the text about the timing of cell culture, saying "48 hours" in the text (page 8) and "72 hours" in figure legend. The method section, page 18, also said 72 hours.
- There is a discrepancy of animal number in the text (Page 8, n=8 per group) and Figure 5C-D (6 mice in the low-dose group and 9 mice in the high-dose group).
- There is a discrepancy when they described KB-AT-23. In the result section, page 6, they stated that KB-AT-23 is a combination of KB-AT-2 and 3, but in the method section, page 17, they said it was from KB-AT-1 and 2.
- The authors should keep consistent using either "hour" or "h" throughout the manuscript.

Referee #3 (Remarks for Author):

It is a well written manuscript with appropriate information.

Though known, will be good to restate the advantages and disadvantages of using antibodies developed in llama.

Some comments:

Overall, the paper can benefit from having some comparators with normal or WT levels. It is important to establish how close we are to the normal, not just show improvement from the disease state.

Table 2: It will be good to see 'normal' plasma sample results for TGA in same table
The ETP with anti-AT agents is going beyond what FVIII can achieve. Normal plasma will help define if this poses a prothrombotic risk?

Figure 2E: TGA assay shows delayed lag time to peak thrombin, and higher AUC as authors acknowledge. Add a reason or hypothesis to explain.

Figure 2E: mark percent AT activity for the green and blue curves

Figure 3: what is the volume of blood loss in WT /non- hemophilia mouse? Is it possible to show a comparison with another treatment option like FVIII? What would this look like if the expt was done at a later time point, 1 hour post antibody? This can help define if the antibodies will only work in acute setting, vs have a longer term effect.

Figure 6: label FIX as FIX ab

Is it possible to give a value to AT inhibition seen *in vivo*? With antibody do we expect to see higher levels of inhibition? A dose dependence with more AAV?

Small grammatical corrections- e.g.,:

- 1) E. coli not E. Coli
- 2) Figure 1: y axis should be thrombin

1st Revision - authors' response

28th Jan 2020

Referee #1 (Remarks for Author):

Recent studies have shown that siRNA-mediated reduction in plasma levels of the anticoagulant antithrombin can successfully be used to restore normal haemostasis in patients with severe haemophilia. In this paper, Barbon et al report an entirely novel approach to treat patients with haemophilia using single-domain antibodies targeting antithrombin. They show for the first time that these nanobodies can correct thrombin generation in haemophilic plasma, and also reverse the bleeding phenotype in murine models of haemophilia A and B. Importantly they further demonstrate

the efficacy of sdAbs to attenuate bleeding phenotype even in the presence of inhibitory antibodies to FIX. This observation is particularly important from the clinical perspective, given the huge morbidity and mortality that continue to be associated with development of high responding inhibitors in patients with severe haemophilia. All together, the paper is well written and follows a logical progression. The data presented are convincing and support the conclusions reached. These novel data are clearly of direct translational relevance, and will undoubtedly be of major interest to a wider readership.

We thank the Reviewer for appreciating the relevance and novelty of our work, and for the suggestions to improve our manuscript.

I have a few minor suggestions that may improve the manuscript:

1. The effect of the sdAbs in down-regulating AT activity is convincing. However, some additional insight into the biological mechanisms through which this effect is achieved would be interesting. For example, it appears that the sdAbs are affecting AT activity rather than causing reduced antigen levels via enhanced clearance. Can different AT activity assays (eg heparin co-factor activity vs progressive activity) provide any additional insights into the mode of action of the sdAbs? Are the effects similar for AT inhibition of its different substrates? Is there any data on the epitopes recognised by the sdAbs ?

We have analyzed the inhibitory activity of KB-AT-23 in chromogenic assays (both in endpoint and progress assays) in the absence and presence of heparin, using thrombin or FXa as enzymes. These data revealed that KB-AT-23 neutralizes antithrombin activity in the absence and presence of heparin for both substrates, albeit that this neutralization is more efficient towards FXa compared to thrombin (See Figs 2B-C). In addition, progress curves indicated that KB-AT-23 resembles a tight binding, competitive inhibitor. This suggests that the sdAb interferes in complex formation between antithrombin and enzyme. Future work will be aimed to elucidate the mechanism of action and the specific epitopes in more detail.

This information is now included in the discussion section (page 13, line 28 – page 14, line 4).

2A. What is the half-life / MRT of recombinant KB-AT-23? To address this question of the Reviewer, we have generated two distinct sdAb-fusion proteins. One consisting of KB-AT-23 fused to a von Willebrand factor (VWF) polypeptide (residues 1261-1478), and one consisting of a bivalent control sdAb (KB-hFX-11), also fused to this VWF polypeptide. These constructs were named KB-AT-23-fus and KB-hFX-11-fus, respectively, and clearance was monitored by using antibodies against the VWF polypeptide. We have chosen to use this approach to avoid that free and antithrombin-bound nanobody are recognized differently by nanobody-directed antibodies. For the clearance experiment, mice were given 10 mg/kg of either protein intravenously, and samples were taken at given time-points (5 min to 24 h) and analyzed for the presence of residual protein. Recoveries at 5 min after intravenous infusion were $93 \pm 18\%$ and $47 \pm 7\%$ ($p=0.012$) for KB-AT-23-fus and KB-hFX-11-fus, respectively. Both proteins were eliminated in a bi-exponential manner. The half-lives of the initial portions (alpha-phase) of the decay curve were 0.3h (95%-CI 0.02-0.6h) and 0.03h (95%-CI 0.02-0.05h) for KB-AT-23-fus and KB-hFX-11-fus, respectively. The half-lives of the terminal portions (beta-phase) were 38 h (95%-CI 21-178h) and 0.7 h (95%-CI 0.5-1.0h) for KB-AT-23-fus and KB-hFX-11-fus, respectively. The notion that KB-AT-23-fus has a substantial longer half-life compared to the similar-sized control protein strongly suggests that the nanobody associates to antithrombin.

These data are now included in Figure 3 and referred to in the results and discussion section (page 7, line 21 – page 8, line 7; page 13, line 3-5)

2B. From a clinical perspective, might be interesting to know whether the inhibitory effect of sdAbs can be overcome by addition of exogenous antithrombin concentrate.

This is an important issue, as having an antidote available for a therapeutic treatment would be relevant in case treatment needs acute reversal. To test the option of antithrombin concentrates, plasma from factor VIII-deficient mice containing KB-AT-23 alone (2 microM) or KB-AT-23 in the presence of antithrombin concentrate (2 microM) were analyzed. In TGT assays, the amount of thrombin generated in FVIII-deficient mouse plasma was 301 ± 133 nM·min, compared to 1085 ± 62 nM·min in plasma of wild-type mice ($p < 0.0001$). In the presence of KB-AT-23, the endogenous thrombin potential (ETP) of FVIII-deficient mouse plasma increased to 910 ± 135 nM·min ($=0.158$ vs wild-type, $p < 0.0001$ versus FVIII-deficient plasma). In the presence of KB-AT-23 and antithrombin concentrate, ETP was markedly decreased (160 ± 18 nM·min) and not

significantly different from FVIII-deficient plasma alone ($p=0.260$). These data demonstrate that antithrombin can be used as antidote to neutralize the activity of the anti-antithrombin nanobody. We have added the thrombin generation data obtained using murine FVIII deficient plasma as new Table 3 to the revised version. Data are referred to in the results and discussion section section (page 7, line 5-14; page 14, line 9-13).

3. In the discussion, the authors comment upon their mechanism of targeting AT (either with recombinant sdAb or with AAV mediated gene transfer) vis a vis that involving siRNA. I think it would be interesting to further expand this discussion - for example in terms of comparing (i) half-lives / duration of effect (ii) reversal potential (iii) respective effects on plasma AT antigen versus activity.

As suggested by the Reviewer, we have adapted the discussion accordingly, thereby including additional data on the half-life of sdAb KB-AT-23 section (page 13, line 9-27; page 14, line 4-8).

4. As noted above, the data demonstrating that the sdAb can ameliorate murine bleeding in the presence of a FIX inhibitor are of direct clinical importance. It would be nice to supplement this with data showing that the sdAB also corrects thrombin generation in human haemophilia plasma in the presence of a high responding inhibitor.

We agree it is of relevance to investigate the effect of inhibitory FVIII antibodies on the capacity of the anti-antithrombin sdAb to ameliorate thrombin generation in FVIII-deficient plasma. To accommodate the requests of both Reviewer 1 (testing in human plasma with inhibitors) and Reviewer 2 (bleeding phenotype in hemophilia A mice with inhibitors), we decided to address this question using plasma of FVIII-deficient mice. The use of murine FVIII-deficient plasma avoids the potential presence of residual FVIII in human hemophilia plasma (which could falsely increase thrombin generation), and reduces the number of animals needed to obtain relevant data.

We analyzed murine FVIII-deficient plasma containing KB-AT-23 (2 microM) in the absence or presence of polyclonal inhibitory anti-FVIII antibodies (8 BU/ml final titer, which blocks >99% FVIII activity) in thrombin generation assays. The amount of thrombin generated (ETP) in FVIII-deficient mouse plasma was 301 ± 133 nM·min, compared to 1085 ± 62 nM·min in plasma of wild-type mice ($p < 0.0001$). In the presence of KB-AT-23, ETP in FVIII-deficient plasma increased to 910 ± 135 nM·min ($p = 0.158$ vs wild-type, $p < 0.0001$ versus FVIII-deficient plasma). In the additional presence of polyclonal anti-FVIII antibodies, ETP remained unchanged (930 ± 43 nM·min; $p = 0.242$ vs wild-type plasma and 0.998 versus KB-AT-23 alone). These data clearly confirm that anti-FVIII inhibitors (which selectively target FVIII) leave the nanobody-targeted antithrombin pathway unaffected.

We have added the thrombin generation data obtained using murine FVIII deficient plasma as new Table 3 to the revised version. Data are referred to in the results section section (page 7, line 5-14).

Referee #2 (Comments on Novelty/Model System for Author):

In this manuscript, Barbon and colleagues reported that using single domain antibody fragments (sdAbs) to inhibit the antithrombin pathway can restore hemostasis in hemophilic mice. They showed that antibody against antithrombin was able to restore thrombin generation in vitro and improved hemostasis in hemophilia A mice. Then they targeted sdAb expression to hepatocytes by AAV and demonstrated the efficacy in hemophilia A mice without inhibitors and hemophilia B mice with inhibitors. The idea is novel. There are some concerns in the manuscript that needs to be addressed.

We would like to thank the Reviewer for the valuable comments to further improve our manuscript.

Concerns

1. They showed (Figure 5F) that animals in the high dose group did get much higher copy number of viral genome per cell as determined by quantitative real-time PCR; however, the phenotypic correction in the high-dose group appears to be not as good as the low-dose group (Figure 5E). How to explain that the animal with 10-fold greater transduction efficacy and better KB-AT-23 expression but blood loss was similar to PBS-treated controls? Is it because animals produced antibodies against viral capsid protein or transduced hepatocytes were attacked by immune cells? Did liver enzyme increase in transduced animals?

The Reviewer raises a valid concern. Figure 5F shows, as expected, a vector dose-dependent increase in vector genome copy number in the liver of the mice treated with the AAV8-hAAT-KB-AT-23 vector. At the phenotypic level, both vector doses result in the correction of the bleeding loss following tail transection (Figure 5E). At both doses, 1×10^{10} and 1×10^{11} vg/kg, blood loss is

significantly reduced compared to untreated mice, and no statistically significant difference in blood loss is measured between the two cohorts of animals treated at the 1×10^{10} and 1×10^{11} vg/kg vector doses. As for the Reviewer's remark that mice with a 10-fold greater transduction efficacy and better KB-AT-23 expression have a blood loss similar to PBS-treated mice, we would like to note that this concerns but 2 of the 5 mice treated with 1×10^{12} vg/mouse, whereas the other three displayed full correction of blood loss. We anticipate that these results originate from the variation that is inherent to the bleeding models that are applied, which is illustrated well in the original publication where this bleeding model was described (Johansen et al. *Haemophilia* 2016, 22:625-631).

As for the possibility that the mice produced antibodies against the viral capsid protein or that transduced hepatocytes were attacked by immune cells, it is important to mention that while immune responses directed against the vector capsid have been frequently observed in humans undergoing AAV gene transfer, these have never been observed in mice or other animal models. Furthermore, when occurring these responses would result in clearance of the transduced hepatocytes thus reducing the vector genome copy number in liver. Thus, we feel that it is a safe assumption to exclude the instance of an immune response directed against transduced hepatocytes in our study. To capture these considerations, we amended the results section of the manuscript as follows:

“In both males and females, we observed a significant amelioration of the bleeding phenotype at both vector doses tested, (Fig. 5E), suggesting that the 1×10^{10} vg/kg was already above the therapeutic threshold for correction of blood loss following TVT. “

We also added wording about defining the minimal effective dose of vector in the discussion section:

“Translating these findings to humans, particularly as a gene therapy, will require further safety and efficacy studies in small and large animal models of hemophilia (Sabatino *et al*, 2012), particularly aimed at defining the minimal effective vector dose that would result in correction of bleeding time.”

Changes are incorporated in the results and discussion section section (page 10, line 4-6; page 15, line 18-21).

2. It is unclear why the manuscript does not include studies in hemophilia A with inhibitors. The authors showed the efficacy of targeting bi-paratopic sdAb expression in hemophilia B mice with inhibitors. They should include studies from hemophilia A with inhibitors to conclude that expression of bi-paratopic sdAb can result in sustained correction of the bleeding phenotype in hemophilic mice with inhibitors.

We understand the point raised by the Reviewer. The choice of the hemophilia B mouse model to test the sdAb efficacy in presence of inhibitors was initially driven by its relevance.

Although the incidence of inhibitors in haemophilia B is much less frequent compared to haemophilia A, treatment options are much more limited. We therefore thought it would be of relevance to demonstrate that our approach could be applied to this rare situation of haemophilia B with inhibitors.

To accommodate the requests of both Reviewer 1 (testing in human plasma with inhibitors) and Reviewer 2 (bleeding phenotype in hemophilia A-mice with inhibitors), we decided to address this question using plasma of FVIII-deficient mice. The use of murine FVIII-deficient plasma avoids the potential presence of residual FVIII in human hemophilia plasma (which could falsely increase thrombin generation), and reduces the number of animals needed to obtain relevant data.

We analyzed murine FVIII-deficient plasma containing KB-AT-23 (2 microM) in the absence or presence of polyclonal inhibitory anti-FVIII antibodies (8 BU/ml final titer, which blocks >99% FVIII activity) in thrombin generation assays. The amount of thrombin generated (ETP) in FVIII-deficient mouse plasma was 301 ± 133 nM·min, compared to 1085 ± 62 nM·min in plasma of wild-type mice ($p < 0.0001$). In the presence of KB-AT-23, ETP in FVIII-deficient plasma increased to 910 ± 135 nM·min ($p = 0.158$ vs wild-type, $p < 0.0001$ versus FVIII-deficient plasma). In the additional presence of polyclonal anti-FVIII antibodies, ETP remained unchanged (930 ± 43 nM·min; $p = 0.242$ vs wild-type plasma and 0.998 versus KB-AT-23 alone). These data clearly confirm that anti-FVIII inhibitors (which selectively target FVIII) leave the nanobody-targeted antithrombin pathway unaffected.

We have added the thrombin generation data obtained using murine FVIII deficient plasma as new Table 3 to the revised version. Data are referred to in the results section section (page 7, line 5-14).

3. They should include data from the WT control group in Figure 3B to show whether infusion of KB-AT-23 was able to completely rescue the bleeding phenotype in hemophilia A mice.

We apologize for having been unclear in the representation of the data. In the original figure 3B, we had indicated the range of blood loss observed in mice that received regular FVIII treatment (n=6; 32.6-245.3 μ l) using a pale grey area limited by dotted lines. These data are in the similar range as those found for wild-type C56Bl6 mice (49-308 μ l; Johansen et al. Haemophilia 2016, 22:625-631). This is now clarified in the legend of the figure.

4. Fig 1B, C, &D, there is a discrepancy between the figure legend and figure shown in Fig B, C, and D. The Figures show bar graphs, but figure legend describes "histogram". What is the N number in those groups?

We thank the Reviewer for pointing to this discrepancy. We have corrected this in the revised legend. Data represent mean \pm SD of three independent experiments.

5. Figure 2B-C, what is the N number in those studies?

Also here the data represent the mean \pm SD of three experiments. This is now added to the revised legend.

6. Figure 3A, in the schematic diagram, it is unclear the relation between \varnothing 2.3 mm and the depth of 0.5 mm.

Also, there is a discrepancy between Figure legend and actual animal number as figure showed 7 data points in the Vehicle group, but the author described 4-6 in the figure legend.

We modified the Figure 3 legend in order to clarify the meaning of \varnothing 2.3 mm and 0.5 mm depth. Also, we corrected the number of animals in the figure legend.

7. In Figure 4B, there is a discrepancy in the figure legend and the text about the timing of cell culture, saying "48 hours" in the text (page 8) and "72 hours" in figure legend. The method section, page 18, also said 72 hours.

We thank the Reviewer for pointing out the typo. We have amended the text (page 9, line 4) with the correct time point.

8. There is a discrepancy of animal number in the text (Page 8, n=8 per group) and Figure 5C-D (6 mice in the low-dose group and 9 mice in the high-dose group).

We modified the text (page 9, line 16) and the figure 5A legend by adding the exact number of animals.

9. There is a discrepancy when they described KB-AT-23. In the result section, page 6, they stated that KB-AT-23 is a combination of KB-AT-2 and 3, but in the method section, page 17, they said it was from KB-AT-1 and 2.

We corrected the materials and methods (page 19, line 23). As described in the results section, the KB-AT-23 is composed of KB-AT-2 and -3.

10. The authors should keep consistent using either "hour" or "h" throughout the manuscript.

We thank the Reviewer for the observation. We amended the text in order to keep consistency and consistently used the term "hour".

Referee #3 (Remarks for Author):

It is a well written manuscript with appropriate information.

We thank the Reviewer for the positive appreciation of our manuscript.

1. Though known, will be good to restate the advantages and disadvantages of using antibodies developed in llama.

As suggested, we have included some of the advantages and disadvantages of llama-derived nanobodies in the text.

2. Overall, the paper can benefit from having some comparators with normal or WT levels. It is important to establish how close we are to the normal, not just show improvement from the disease state.

In figures 3B and 5E, the range of blood loss after a tail-vein transection observed in FVIII-treated mice (32.6-245.3 μ l) is now indicated using a pale grey area limited by dotted lines. In Fig 6F, data from wild-type mice are added to the figure. In addition, data from F9^{-/-} mice treated with AAV-vectors encoding human FIX are now included (pale grey area limited by dotted lines).

3. Table 2: It will be good to see 'normal' plasma sample results for TGA in same table. The ETP with anti-AT agents is going beyond what FVIII can achieve. Normal plasma will help define if this poses a prothrombotic risk?

In the revised version, we have now added data for normal plasma (see Table 2). As the Reviewer will appreciate, thrombin generation in the presence of the anti-antithrombin sdAbs, thrombin generation exceeds that of normal human plasma. A more in-depth description of how to interpret these thrombin generation is provided below, under point 4 in our response to this Reviewer.

4. Figure 2E: TGA assay shows delayed lag time to peak thrombin, and higher AUC as authors acknowledge. Add a reason or hypothesis to explain.

The thrombin generation curve represents a mixture of different, overlapping steps. The initial phase needed to reach the thrombin peak, is very much dependent on FIXa/FVIIIa activity, and is limited by Tissue Factor Pathway Inhibitor (TFPI)-mediated inhibition of FXa/FVIIa/TF. Once the thrombin peak has been reached, down-regulation is initiated, depending on inactivation of the labile FVIIIa cofactor and inhibition of factor Xa and thrombin by antithrombin. Since the FIXa/FVIIIa complex is missing, the time to reach the thrombin peak is delayed. Furthermore, sdAb-mediated neutralization of antithrombin will result in prolonged action of thrombin and FXa, manifested by a slower return to baseline in the thrombin generation curve. It should be noted that the thrombin generation assay provides only a partial representation of the pro- and anticoagulant pathways that are available *in vivo*, which is particularly relevant in case coagulation proceeds independently of the FIXa/FVIIIa-complex. Importantly, the thrombomodulin-dependent APC-pathway that mediates inactivation of FVa is markedly underrepresented in this thrombin generation assay. It seems therefore, that *in vitro* thrombin generation in FVIII- or FIX-deficient plasma in the presence of an antithrombin-inhibitor results in artificially exaggerated thrombin generation. Indeed, our *in vivo* data are in support of this hypothesis, since the presence of anti-antithrombin sdAb results in near-normalization of the bleeding tendency, while D-dimer levels, a marker for thrombosis, were not increased.

These considerations are now added to the discussion section of the revised manuscript (page 14, line 14-25).

5. Figure 2E: mark percent AT activity for the green and blue curves

Residual activity of antithrombin has been measured using FXa-dependent activity assays, as thrombin-based assay have a tendency to over-estimate antithrombin activity (Beek et al. Blood Coagul Fibrinolysis 2000, 11:127-135). Based on this assay, a concentration of 0.2 mg/ml results in 81.1 \pm 6.1% antithrombin inhibition, and 0.1 mg/ml results in 54.1 \pm 8.4% antithrombin inhibition. This is now mentioned in the legend of the figure.

6. Figure 3: what is the volume of blood loss in WT /non- hemophilia mouse? Is it possible to show a comparison with another treatment option like FVIII? What would this look like if the expt was done at a later time point, 1 hour post antibody? This can help define if the antibodies will only work in acute setting, vs have a longer term effect.

We apologize for having been unclear in the representation of the data. In the original figure 3B, we had indicated the range of blood loss observed in mice that received regular FVIII treatment (n=6; 32.6-245.3 μ l). These data are in the similar range as those found for wild-type C56Bl6 mice (49-308 μ l; Johansen et al. Haemophilia 2016, 22:625-631). This is now clarified in the legend of the figure. We have also added data of mice treated with recombinant FVIIa (1 mg/kg), which show a full correction of the bleeding tendency. Interestingly, FVIIa and KB-AT-23 were not significantly different.

As for an infusion given 1 hour before the tail vein transection rather than 10 min, we know from our clearance experiments, that 1 hour after infusion sdAb levels are reduced from 186 microgram/ml to 125 microgram/ml, and that antithrombin activity increases from 20% to 36%. We anticipate that this may result in a slightly increased bleeding tendency. In its current form, sdAb

KB-AT-23 (when given intravenously) would be more appropriate for acute treatment rather than prophylactic treatment. It is relevant to note, that variants of KB-AT-23 are under consideration that will allow a longer half-life and subcutaneous application. Such molecules would indeed allow for prophylactic treatment.

7a. Figure 6: label FIX as FIX ab

As suggested by the Reviewer, we have adapted the labeling of the panels, by changing FIX into FIX-Ab.

7b. Is it possible to give a value to AT inhibition seen in vivo? With antibody do we expect to see higher levels of inhibition?

In figure 6E, we have indicated residual antithrombin activities following treatment with PBS or AAV8-hAAT-KB-AT-23 in FIX deficient mice in the absence or presence of anti-FIX inhibitory antibodies. There is a significant reduction in antithrombin activity in mice treated with AAV8-hAAT-KB-AT-23 compared to PBS-treated mice, both at 4 and 8 weeks. However, there is no difference in antithrombin inhibition between mice with or without anti-FIX inhibitors. This is compatible with the fact that the antibodies are specific to FIX, and do not cross-react with antithrombin. The fact that anti-FIX or anti-FVIII antibodies leave the action of anti-antithrombin sdAbs unaffected is further illustrated by our experiments showing that KB-AT-23 has a similar effect on thrombin generation in the absence and presence of anti-FVIII antibodies (see new Table 3).

7c. A dose dependence with more AAV?

The administration of different vector doses allows to modulate the sdAb plasma concentration that can be achieved over time (as shown in Fig.5B), as well as the usage of different promoters can modulate expression levels. While it is expected that increasing sdAb plasma concentration would result in a dose-dependent lowering of AT activity and to what extent, future studies in small and large animal models of haemophilia will help to identify the minimal effective dose that restores coagulation.

8. Small grammatical corrections- e.g.,:

- 1) E. coli not E. Coli
- 2) Figure 1: y axis should be thrombin

We thank the Reviewer for pointing out these errors, which have been corrected in the revised version.

2nd Editorial Decision

6th Feb 2020

Thank you for the submission of your revised manuscript to EMBO Molecular Medicine. We have now received the enclosed reports from the 2 referees who accepted to review your revised manuscript. As you will see, they are supportive of publication, and I am thus pleased to inform you that we will be able to accept your manuscript pending the following final editorial amendments.

***** Reviewer's comments *****

Referee #1 (Remarks for Author):

I am grateful for the additional work carried out by the authors.
I have nothing further.

Referee #2 (Remarks for Author):
[suitable for publication]

The authors performed the requested editorial changes.

Corresponding Author Name: Peter Lenting & Federico Mingozzi

Manuscript Number: EMM-2019-11298